

# Direct integration of reservoirs' operations in a hydrological model for streamflow estimation: coupling a CLSTM model with MOHID-Land

Ana R. Oliveira, Tiago B. Ramos, Lígia Pinto, Ramiro Neves

Centro de Ciência e Tecnologia do Ambiente e do Mar (MARETEC-LARSyS), Instituto Superior Técnico, Universidade de Lisboa, Av. Rovisco Pais, 1, 1049-001 Lisboa, Portugal

*Correspondence to*: Ana R. Oliveira (anaramosoliveira@tecnico.ulisboa.pt)

**Abstract.** Knowledge about streamflow regimes and values is essential for different activities and situations, in which justified decisions must be made. However, streamflow behavior is commonly assumed as non-linear, being controlled by

various mechanisms that act on different temporal and spatial scales, making its estimate challenging. An example is the construction and operation of infrastructures such as dams and reservoirs in rivers. The challenges faced by modelers to correctly describe the impact of dams on hydrological systems are considerable. In this study, an already implemented, calibrated, and validated solution of MOHID-Land model for natural regime flow in Ulla River basin was considered as baseline. The referred watershed comprehends three reservoirs. Outflow values were estimated considering a basic operation

rule for two of them (run-of-the-river dams) and considering a data-driven model of Convolutional Long Short-Term Memory (CLSTM) type for the other (high-capacity dam). The outflow values obtained with the CLSTM model were imposed in the hydrological model, while the hydrological model fed the CLSTM model with the level and the inflow of the reservoir. This coupled system was daily evaluated in two hydrometric stations located downstream of the reservoirs, resulting in an improved performance compared with the baseline application. The analysis of the modelled values with and

without reservoirs further demonstrated that considering dams' operations in the hydrological model resulted in an increase of the streamflow during the dry season and a decrease during the wet season but with no differences in the average streamflow. The coupled system is thus a promising solution for improving streamflow estimates in modified rivers.

## 1 Introduction

Knowledge about streamflow, including water quantity and quality, is fundamental for monitoring and controlling the

environmental impacts of several activities and situations, including infrastructures design, support in decision-making processes, irrigation scheduling, design and implementation of water management systems, environmental management, studies of river and watershed behavior, flood warning control, optimal water resources allocation, prediction of droughts, and management of reservoir operations (Mehdizadeh et al., 2019; Mohammadi et al., 2021; Hu et al., 2021). However, the task of delivering information about streamflow can be challenging since it commonly assumes a non-linear behavior, being



controlled by various mechanisms that act on different temporal and spatial scales (Wang et al., 2006). These non-linear forcing mechanisms include meteorological conditions, land use, infiltration, morphological features of the river, and catchment characteristics (Mohammadi et al., 2021). The complex and laborious process of streamflow estimation is usually exacerbated when the natural regime flow is modified by anthropogenic activities and human decisions. In this sense, reservoirs are a major concern in hydrological modelling since most models are not prepared to directly consider the

existence of such infrastructures and the resulting alterations caused on the natural regime flow by their operations (Dang et al., 2020). If hydrological models are prepared to study and comprehend the behavior of natural systems, the lack of information about reservoirs' operations such as operating rules and flood contingency plans makes it impeditive for a correct representation of those infrastructures.

As pointed out by Dang et al. (2020), a postprocess methodology is often used to impose reservoirs' operations on

hydrologic-hydraulic models. This way, the need for modifying models' structures is avoided. However, Bellin et al. (2016) considered the direct representation of reservoirs water storage and operation as the best approach to correctly simulate such systems. Nevertheless, the challenges faced   are many, having limited the number of studies carried out (Dang et al., 2020). Recently, Xiong et al. (2019) developed a statistical framework where an indicator combining the effects of reservoir storage capacity and the volume of the multiday antecedent rainfall input was used to assess the impact of a reservoir system on

flood frequency and magnitude in downstream areas of the Han River, China. Yun et al. (2020) modified the structure of the Variable Infiltration Capacity (VIC) model to include a reservoir module for estimating the variation of streamflow and flood characteristics when impacted by climate change and reservoir operation in the Lancang-Mekong River basin, Southeast Asia. Also using a modified VIC model, Dang et al. (2020) simulated storage dynamics of water reservoirs again in the Lancang-Mekong River basin. In both studies, a comparison between the model results with and without reservoirs

was provided. It is important to denote that both Yun et al. (2020) and Dang et al. (2020) imposed operation rules on the model, with the former authors giving more importance to flood control and environmental protection while the latter focused on energy production. Also, Hughes et al. (2021) used a modified version of the SHETRAN model to simulate the streamflow considering the influence of reservoirs in Upper Cocker catchment, United Kingdom. The authors considered a weir model and two tests were performed: first the weir was simulated as static (with closed sluice) to identify the sluice

operating rules by comparing results with the known outflow timeseries; second the weir model was run as non-static to implement the sluice operating rules deducted from the first approach. All studies mentioned above reproduced reservoirs' behavior considering their operation rules, which in most cases are difficult to obtain or are very laborious to reproduce.

The application of operation rules may often be adapted to specific conditions, objectives, or constraints based on the knowledge and experience of operators (Yang et al., 2019). This makes the reservoirs' operation deviate from the reference

operation curves, invalidating the sole use of physical-based models and the use of pre-established rule curves to reproduce the reservoir behavior in real-time. To overcome this issue, Yang et al. (2019) referred that machine learning methods, with their capacity to understand, extract, and reproduce complex high-dimensional relationships, can be an efficient and easy-to-use solution to reproduce reservoirs' operations, contemplating the reference operation rules as well as the operators'



historical experience. In this sense, the referred authors used recurrent neural network (RNN) models to extract reservoirs'
operation rules from the historical operation data of three multipurpose reservoirs located in the upper Chao Phraya River
basin, Thailand. Also considering the use of the Geomorphology-based hydrological model (GBHM) to forecast the
reservoir's inflow, Yang et al. (2019) achieved a real-time reservoir outflow forecast. Dong et al. (2023) proposed a similar
approach to improve the reconstruction of daily streamflow timeseries in the Upper Yangtze River Basin, China. These
authors proposed a practical framework to quantitatively assess the cumulative impacts of reservoirs' operation on the
hydrologic regime, coupling two data-driven models, namely an extreme gradient boosting (XGBoost) model and an
artificial neural network (ANN) model, with a high-resolution hydrologic model, and following a calibration free conceptual
reservoir operation scheme. The data-driven models were used to predict the outflow of reservoirs with historical operation
data, while the calibration-free conceptual reservoir approach was used to simulate the outflow in data limited reservoirs.
The study presented by Dong et al. (2023) is a rare example of a promising solution for improving streamflow prediction in
highly modified catchments, which this study aims to follow.

In the present study, the physical-based, distributed MOHID-Land model (Oliveira et al., 2020) was coupled with a
Convolutional Long Short-Term Memory (CLSTM) model to estimate the daily outflow in Portodemouros reservoir,
Galicia, Spain. The results obtained with the CLSTM model were estimated considering the reservoir's level and inflow
simulated by MOHID-Land and then imposed in that same model for streamflow simulation downstream the reservoirs.
However, the CLSTM model was first trained and tested using historical data. Thus, the main aim of this study is to verify
the capacity of the coupled system to improve streamflow estimation downstream Portodemouros reservoir. This study
demonstrates the ability of the proposed approach to directly simulate reservoirs' operations in a hydrological simulation and
validates a solution that is accessible and easy to implement.

## 2 Materials and methods

### 2.1 Description of the study area

The Ulla River watershed is located in the Galicia region, Northwest of Spain, and drains an area of 2803 km$^2$ discharging on
Ria de Arousa estuary (Figure 1). Ria de Arousa is one of the most important coastal water bodies in Galicia, having the Ulla
and Umia rivers as major tributaries, and mainly used for recreative and fishery activities (da Silva et al., 2005; Outeiro et
al., 2018; Blanco-Chao et al., 2020; Cloux et al., 2022). The maximum and minimum elevations of the Ulla watershed are
1160 m and -0.75 m, respectively, and the main watercourse has a bed length of 142 km with its source at an altitude of 600
m. The watershed is inserted into an area characterized by a warm-summer Mediterranean climate (Csb) according to
Köppen-Geiger classification. The annual precipitation is about 1100 mm, with rainy months from October to May. The
annual average temperature is 12°C, reaching a maximum of 18°C in August and a minimum of 7°C in February. According
to Nachtergaele et al. (2009), the main soil units in the Ulla river watershed are Umbric Leptosols and Umbric Regosols,



representing 69% and 31%, respectively. The main land uses are forest, occupying 57.2% of the area, and semi natural and agricultural areas, covering 40.3% (CLC 2012, n.d.).

**Figure 1 Ulla River watershed location, digital terrain model, and identification of hydrometric stations and reservoirs.**

There are three reservoirs in the watershed: Portodemouros, Bandariz, and Touro (Figure 1). Those reservoirs were constructed in cascade and work collectively, with Portodemouros placed at the beginning of the cascade, Touro at the end, and Bandariz in between. Portodemouros has a total capacity of 297 hm$^3$, while Bandariz and Touro present much lower capacities, totalizing 2.7 hm$^3$ and 3.78 hm$^3$, respectively. Due to its significative storing capacity, Portodemouros reservoir can be used for flood control, however, the set of reservoirs is mainly responsible for energy production. The patterns of daily inflow and outflow of the two last reservoirs are very similar, since they are run-of-the-river dams (Figure 2b and c). However, Portodemouros works in a different way, presenting significative differences between the inflow and outflow patterns (Figure 2a and d).

**Figure 2 Comparison of inflow and outflow in (a) Portodemouros, (b) Touro, and (c) Bandariz reservoirs for the period 2010-2018, and in (d) Portodemouros reservoir for the period 1990-2018.**

### 2.2 MOHID-Land description

MOHID-Land is an open-source model (https://github.com/Mohid-Water-Modelling-System/Mohid) and is part of the MOHID (Hydrodynamic Model) Water Modelling System. It is a fully distributed and physically based model adopting mass and momentum conservation equations considering a finite volume approach (Trancoso et al. 2009, Canuto et al., 2019, Oliveira et al., 2020). The model estimates water fluxes between four main compartments, namely, the atmosphere, the soil surface, the river network, and the porous media, which is also intimately related with the vegetation compartment. Excepting the atmosphere compartment, which is only responsible for providing the meteorological data needed to impose surface boundary conditions, the processes in all the other compartments are explicitly simulated.

In MOHID-Land, the atmosphere compartment can deal with space and/or time variable data, and the input properties include precipitation, air temperature, relative humidity, wind velocity, and solar radiation and/or cloud cover.

The simulated domain is discretized considering two grids, one in the surface plane and other in the vertical direction. While the first is defined according to the coordinate system chosen by the user, the last follows a cartesian coordinate system. The surface water movement is computed considering a 2D surface grid and solving the Saint-Venant equation in its conservative form, accounting for advection, pressure, and friction forces. The Saint-Venant equation is also solved one-dimensionally (1D) for the river network. This network is derived from the digital terrain model represented in the 2D surface grid by connecting surface cell centers (nodes) and is characterized by a cross-section geometry defined by the user. The water fluxes between these two (2D and 1D) compartments are estimated according to a kinematic approach, neglecting bottom friction, and using an implicit algorithm to avoid instabilities.

The porous media is discretized by combining the 2D surface grid with the vertical cartesian grid, defining a 3D domain with variable thickness layers. This compartment can receive or lose water from the river network, with fluxes being computed





considering a pressure gradient in the interface of these two mediums. Besides the water coming from the drainage network, the porous media also receives water from the infiltration process, which is calculated according to Darcy's law. In this 3D domain, the water movement is simulated using the Richards equation and considering the same grid for saturated and unsaturated flow. The soil hydraulic parameters are described using the van Genuchten-Mualem functional relationships (Mualem, 1976; van Genuchten, 1980). The saturation is reached when a cell exceeds the soil moisture threshold value defined by the user and, in that case, the model considers the saturated conductivity to compute flow, with pressure becoming hydrostatic and corrected by friction. To compute the lateral flow, the horizontal saturated hydraulic conductivity is given by the product of the vertical saturated hydraulic conductivity ($K_{sat,ver}$) and a factor ($f_h$) set by the user.

The soil water loss is mainly due to the evapotranspiration process, which is computed taking into account weather, crop, and soil conditions. The reference evapotranspiration ($ET_o$) is first computed according to the FAO Penman–Monteith method (Allen et al., 1998). Then, the potential crop evapotranspiration ($ET_c$) is obtained by multiplying the $ET_o$ by a single crop coefficient ($K_c$) representing standard crop conditions. $ET_c$ values are then partitioned into potential soil evaporation and crop transpiration rates based on the leaf area index (LAI) following Ritchie (1972). LAI is simulated using a modified version of the EPIC model (Neitsch et al., 2011, Williams et al., 1989) and considering a heat units approach for crop development, the crop development stages, and crop stress (Ramos et al., 2017). The actual transpiration is calculated based on the macroscopic approach proposed by Feddes et al. (1978), where root water uptake reductions are estimated considering the presence of depth-varying stressors (Šimůnek and Hopmans, 2009, Skaggs et al., 2006). The actual soil evaporation is estimated from the potential soil evaporation by imposing a pressure head threshold value (ASCE, 1996).

To avoid instability problems and save computational time, the model allows the use of a variable time step, which reaches higher values during dry seasons and lower values in rainy periods when water fluxes increase.

### 2.2.1 Reservoirs module

Besides the main modules described above, MOHID-Land can also consider the existence of reservoirs in the river network domain. The operation of a reservoir needs several characteristics to be defined, namely, the minimum and maximum volumes, the minimum outflow (the definition of the maximum outflow is optional), the curve defining the relation between the level and the stored volume, the type of operation, the location in terms of coordinates, and the identification of the node in the river network where the reservoir is placed. Reservoir's operation may be defined by the relationship between the level and the outflow as absolute value or as a percentage of the inflow, the percentage of the stored volume and the outflow as absolute value or as a percentage of the inflow, and the percentage of the stored volume and the outflow as a percentage of the maximum outflow. The user can also define the existence of discharges (in and/or out) and the state of the storage capacity (full, filled with a percentage of the total capacity, or empty) at the beginning of the simulation. In that sense, the reservoirs module works with each reservoir as a box where a mass balance is performed. This mass balance takes into account the stored volume and the amount of water that enters and leaves the reservoir. The former considers the inflow from the river network and any input discharge defined by the user. The latter considers the outflow estimated by the type of



operation and any output discharge defined by the user. The new stored volume is transformed into a level according to the level/volume curve specified by the user.

### 2.2.2 Model set-up

The MOHID-Land model was already implemented, calibrated, and validated in the study area as detailed in Oliveira et al.
(2020). This study was carried out from 01/01/2008 to 31/12/2017. Only the natural regime flow in the watershed was considered, with model calibration and validation using data from hydrometric stations not influenced by reservoirs' operations. Following the sensitivity analysis performed, the best solution for the Ulla River model implementation was obtained considering a constant quadrangular horizontally spaced grid with 215 columns (West-East direction) and 115 rows (North-South direction), and a resolution of 0.005° (~500 m). The calibrated parameters were the $K_{sat,ver}$, the $f_h$ factor, and
the dimensions of the cross-sections in the river network.

The elevation of the calibrated solution was interpolated based on the digital terrain model from the European Environment Agency (European Digital Elevation Model (EU-DEM), n.d.), which has a resolution of 0.00028° (~30 m). The Manning coefficient for the river network was set to 0.035 s m$^{-1/3}$, and the river cross-sections were assumed as rectangular with the dimensions varying according to the drained area of each node (Table 1).
**Table 1 Cross-sections dimensions.**

The surface Manning coefficients were specified based on the CLC 2012 (CLC 2012, n.d.) data. For each land use, a Manning coefficient was first defined according to Pestana et al. (2013). Considering the interpolation process, those values varied from 0.023 to 0.298 s m$^{-1/3}$. CLC 2012 data was further used to identify the vegetation in the watershed, which were made to correspond to data (vegetation growth parameters) in the MOHID's vegetation database. For each type of
vegetation, a single crop coefficient ($K_c$) was adopted based on Allen et al. (1998) tabulated values. After the interpolation process, the $K_c$ values varied from 0.15 to 1.0.

The soil domain was vertically discretized considering three horizons that comprehended six grid layers. The layers had variable thickness increasing from surface to bottom: 0.3 (surface), 0.3, 0.7, 0.7, 1.5, and 1.5 m (bottom). The first horizon included the first two layers, while the second horizon included the two middle layers, and, finally, the bottom horizon
considered the last two layers. The van Genuchten-Mualem soil hydraulic parameters were obtained from the multilayered European Soil Hydraulic Database (ESHD, Tóth et al., 2017). For the surface horizon, ESHD data at 0.3 m depth was used to represent soil hydraulic data; ESHD data at 1.0 m depth was used to characterize the middle horizon; ESHD data at 2.0 m depth described the bottom horizon. In each of these horizons, three different sets of soil hydraulic data were identified (Figure 3). After model's calibration, the van Genuchten-Mualem soil hydraulic parameters assumed the values presented in
Table 2 for each set. The horizontal saturated hydraulic conductivity was obtained assuming the $f_h$ equal to 10.

**Figure 3 Soil IDs for each horizon: (a) surface; (b) middle; and (c) bottom horizons.**

**Table 2 Soil hydraulic properties by soil ID: θs – saturated water content; θr – residual water content; η and α – empirical shape parameters; Ksat,ver – vertical saturated hydraulic conductivity; and l – pore connectivity/tortuosity parameter.**



The meteorological boundary conditions were extracted from the ERA5-Reanalysis dataset (Hersbach et al., 2017), which is

a gridded product with a resolution of 0.28125° (~31 km) and makes available meteorological variables with an hourly time

step. The variables used and interpolated to the case study grid were the u and v components of wind velocity at 10 m height,

dewpoint and air temperatures at 2 m height, surface solar radiation downwards, surface pressure, total cloud cover, and total

precipitation.

**Reservoirs set-up**

The three reservoirs in the studied watershed were implemented according to the characteristics presented in Table 3. Their

curves relating the level and the stored volume are given in Figure 4. These data were made available by Augas de Galicia

(Augas de Galicia, 2022), which is a public entity managing the Galicia-Costa basin district.

**Table 3 Implemented characteristics for Portodemouros, Bandariz and Touro reservoirs.**

**Figure 4 Level/stored volume curves for (a) Portodemouros, (b) Bandariz, and (c) Touro reservoirs.**

The operation for Bandariz and Touro reservoirs was defined based on the relation between the percentage of the stored

volume and the outflow as a percentage of the inflow. If the stored volume was between 0 and 95%, the reservoir had no

outflow. If the stored volume was above 96%, the outflow equaled the inflow, i.e., all the amount of water that entered the

reservoir each instant left the reservoir in the same instant. For Portodemouros, no operation rule was set since there was no

clear relation between the inflow and outflow values to be used in MOHID-Land. Thus, the daily outflow of Portodemouros

reservoir was estimated using a neural network model and imposed to the hydrologic model as a timeseries. Additionally,

and by default, if the stored volume of any reservoir was equal or above the total capacity, the amount of water that reached

the reservoir is transformed into outflow.

## 2.3 Neural network model for reservoir outflow estimation

To estimate Portodemouros reservoir daily outflow, a neural network model was developed and tuned. It was composed by a

combination of convolutional and a long short-term memory layers, hereafter defined as convolutional long short-term

memory (CLSTM) model. This type of model was already applied for streamflow estimation by Ni et al. (2020) and Ghimire

et al. (2021), who reported that, when compared with other neural network models (convolutional neural network, long

short-term memory, multi-layer perceptron, extreme learning machine, etc.), the CLSTM represented the best solution. The

demonstrated good behavior of CLSTM models is mainly related to its structure, which begins with the use of convolutional

layers, responsible for the extraction of patterns in the input variables, and follows with long short-term layers, which are

responsible for the prediction itself.

As referred by Wang et al. (2019), convolutional neural networks (CNN) have their origin in artificial neural networks

(ANN) but instead of fully connected layers, CNN use local connections, giving more importance to high correlations with

nearby data. Developed by LeCun and Bengio (1995) to identify handwritten digits, CNN uses convolutional filtering to

achieve high correlation with neighboring data. This means that this type of network works based on weight sharing concept,





with the filters' coefficients being shared for all input positions and their number and values being essential to capture data patterns (Wang et al., 2019, Barino et al., 2020, Chong et al., 2020). CNNs are thus recognized as more suitable solutions to identify local patterns, with a certain identified pattern being able to be recognized in another independent occurrence (Tao et al., 2019). As Ghimire et al. (2021) describes, CNN models can be used to identify patterns in one (1D), two (2D) or three (3D) dimensions. Being more adequate for time series data analysis, the 1D CNN solution was selected to be used in this study as input layer. This selection avoided the manual feature extraction process since 1D convolutional algorithms are known for their powerful capability of doing it automatically. According to Huang et al. (2020), the time needed for training CNN models is one of its main weaknesses.

As a type of recurrent neural network (RNN) model, long short-term memory (LSTM) models are known for their capacity to maintain historical information about all the past events of a sequence using a recurrent hidden unit (Elman, 1990, LeCun et al., 2015, Lipton et al., 2015). This characteristic makes RNN very suitable for time series data modelling (Bengio et al., 1994, Hochreiter and Schmidhuber, 1997, Saon and Picheny, 2017). However, RNN models demonstrate inability in learning long-distance information because of their already known vanishing and exploding gradient problems during the training process (Ghimire et al., 2021). Trying to solve this RNN problem, Hochreiter and Schmidhuber (1997) developed the LSTM structure, which has the capacity to learn long-term dependencies (Xu et al., 2020).

### 2.3.1 Input data

The forcing variables were selected from a set that included the daily values of inflow, level, precipitation, temperature, and volume. The usage of the outflow values as a forcing variable was avoided because, when there are no observed values, the outflow data generated by the model must be used to feed the model itself, which can lead to an accumulation and propagation of errors in the estimated values. Several tests were performed considering different forcing variables and their combinations to verify which better estimate the daily outflow from Portodemouros reservoir. Also, different time lags of those forcing variables were tested. The analysis of the tests results shows that the best performance of CLSTM model was obtained with inflow and level used as forcing variables, both considering the values of 1, 2 and 3 days before the forecasted day.

The daily values of inflow, level, and volume were provided by Augas de Galicia, and original hourly values of precipitation and temperature were obtained from ERA5-Reanalysis dataset, being then accumulated or averaged considering a daily time step. The dataset made available by Augas de Galicia covered a period of about 29 years, with data from 01/01/1990 to 16/07/2018.

### 2.3.2 Structure

In this study, the model structure was developed based on Python language and using Keras package (Chollet et al., 2015), on top of TensorFlow (Abadi et al., 2016). As referred before, a CLSTM model is based on convolutional and long short-term memory layers. The types of layers made available by Keras package and used here were the Conv1D, MaxPooling,



LSTM and dense. After several tests, the adopted model's structure included a Conv1D input layer followed by a MaxPooling layer. Then, two other sets of Conv1D plus MaxPooling layers were adopted. After those, an LSTM layer was
introduced, and the output layer was selected to be a dense layer (Figure 5).

**Figure 5 CLSTM structure.**

For convolutional layers no activation function was defined, while the LSTM layer was activated with the hyperbolic tangent function. For the output dense layer, the exponential linear unit function was used as activation function.

The optimizer, i.e., the training algorithm, was selected to be the Nadam algorithm, with a learning rate of $1\times10^{-3}$, and an
epsilon value of $1\times10^{-7}$. The loss was estimated using the mean absolute error. Finally, the number of epochs and the batch size were also a target for some tests, with adopted values of 300 and 20, respectively.

### 2.3.3 Model optimization

The model optimization considered two phases, namely, the manual tunning of hyperparameters, and the optimization of weights reached with the training process. In both cases, the structure presented above was exposed to a subset of the
original dataset, i.e., the training dataset, where the forcing and target variables were included. The training dataset was handled and prepared with Pandas (McKinney, 2010) and Scikit-learn (Pedregosa et al., 2011) packages, with the data being delayed with the first and scaled with the latter. The scaling function was the "MinMaxScaler", which applies equation 1 to each variable in the dataset independently:

$$x_{scaled} = \frac{(x-x_{min})}{x_{max}-x_{min}}(M-m)+m,\tag{1}$$

where $x_{scaled}$ is the scaled value, x is the original value, $x_{max}$ and $x_{min}$ are the maximum and minimum values of the variable being scaled, and M and m are the maximum and minimum values of the desired range of the scaled data. Considering that the maximum values of the variables cannot be represented in the dataset, the desired range was defined from 0 to 0.9.

The tunning process was carried out to optimize the hyperparameters of the model. Several values for the number of filters and the kernel size for convolutional layers, and the number of units for the LSTM layer were tested. The best performance
was reached with 16 filters and a kernel size of 10 for all the three convolutional layers and 10 units for the LSTM layer. The pool size was set as 2 for the first and second MaxPooling layers, and as 1 for the third layer of this type.

The training process consists of changing the weights and bias values of a model to improve its capacity to estimate the target variable. The initialization of those values followed the default definitions of Keras package for all the layers, which means that the weights were initialized according to the Glorot uniform method (Glorot and Bengio, 2010), and the bias
were initialized with value 0. However, this type of initialization and the consequent training process have a random nature associated, repeatedly resulting in different estimations of the same target variable even considering the same forcing variables and the same trained structure. To overcome this problem, the CLSTM model was exposed, trained, and the final weights were saved 100 times always considering the same training dataset, with the results being evaluated individually for





each experiment. Based on these results, the model with the best performance was selected to estimate the outflow values for
Portodemouros reservoir.

## 2.4 Coupling MOHID-Land and CLSTM models

The operationality of the coupled system, which includes the CLSTM and MOHID-Land simulations, was divided into two
phases, one that comprehended the warm-up period, and other including the calibration and validation periods defined in
Oliveira et al. (2020). Since MOHID-Land is a physical model, it was necessary to consider an initial warm-up period for the
stabilization of the hydrological processes and to avoid the influence of the errors related to the imposed initial conditions in
the results.

In both phases, models were simulated on a daily basis, taking advantage of the possibility of doing continuous simulations
in MOHID-Land. This means that in every simulation, the state of the system in the last simulated instant is saved and can be
used as the initial state in the next simulation if date and time match.

In the warm-up simulation, used to stabilize the hydrological model, the  reservoirs' module was deactivated. In the end of
the warm-up period, the reservoirs' module was activated and the initial conditions (level and stored volume) for the three
reservoirs were manually imposed considering the measured values. Then, for each simulated day, the CLSTM model was
the first to be run. This model was loaded with the weights already optimized and received the information about the level
and the inflow of Portodemouros reservoir estimated by MOHID-Land for the three days before the simulated day. The
CLSTM used this information to estimate the outflow for the simulated day. The outflow value estimated by the CLSTM
model was then imposed in MOHID-Land. A scheme representing the described process to couple both models is presented
in Figure 6. This scheme was coded in the Python language.

**Figure 6 Operationality scheme for the modelling process.**

## 2.5 Model's evaluation

The CLSTM model used to predict the outflow from Portodemouros reservoir was evaluated considering a subset of the
original dataset from Augas de Galicia, which was not previously exposed to the trained model. That subset is known as test
dataset and contained pairs of forcing (inflow and level) and target (outflow) variables. Thus, the outflow was estimated
based on the forcing variables and was then compared to the corresponding measured outflow. This comparison was based
on a visual analysis, and the estimation of four different statistical indicators, namely, the coefficient of determination ($R^2$),
the percentage bias (PBIAS), the ratio of the root mean square error to the standard deviation of observation (RSR), and the
Nash-Sutcliffe modeling efficiency (NSE), which were computed as follows:

$$R^2 = \left[\frac{\Sigma_{i=1}^{p}\left(X_i^{obs}-X_{mean}^{obs}\right)\left(X_i^{sim}-X_{mean}^{sim}\right)}{\sqrt{\Sigma_{i=1}^{p}\left(X_i^{obs}-X_{mean}^{obs}\right)^2}\sqrt{\Sigma_{i=1}^{p}\left(X_i^{sim}-X_{mean}^{sim}\right)^2}}\right]^2,  \qquad (2)$$





$$PBIAS = \frac{\sum_{i=1}^{p}\left(X_i^{obs} - X_i^{sim}\right)}{\sum_{i=1}^{p} X_i^{obs}} \times 100, \tag{3}$$

$$RSR = \frac{RMSE}{STDEV_{obs}} = \frac{\sqrt{\sum_{i=1}^{p}\left(X_i^{obs} - X_i^{sim}\right)^2}}{\sqrt{\sum_{i=1}^{p}\left(X_i^{obs} - x_{mean}^{obs}\right)^2}}, \tag{4}$$

$$NSE = 1 - \frac{\sum_{i=1}^{p}\left(X_i^{obs} - X_i^{sim}\right)^2}{\sum_{i=1}^{p}\left(X_i^{obs} - x_{mean}^{obs}\right)^2}, \tag{5}$$

where $x_i^{obs}$ and $x_i^{sim}$ are the outflow values observed and estimated by the model on day i, respectively, $X_{mean}^{obs}$ and $X_{mean}^{sim}$ are the average outflow considering the observed and the modelled values in the analyzed period, and p is the total number of days/values in this period. The test dataset corresponded to 10% of the size of the original dataset and covered the period between 19/09/2015 and 16/07/2018, totalizing 1023 daily values.

In this study the evaluation of streamflow values focused the hydrometric stations placed downstream the set of reservoirs and intended to verify the behavior of the coupled modelling system (MOHID-Land+CLSTM). This evaluation was performed by comparing the streamflow values estimated by the coupled modelling system with those measured in Ulla-Touro and Ulla-Teo hydrometric stations. The validation of the coupled system was made from 01/01/2009 to 31/12/2017 and was based on a visual analysis and the four statistical indicators presented before, namely, the $R^2$, PBIAS, NSE, and

RSR. According to Moriasi et al., 2007, the NSE and the R2 values should be higher than 0.5 and the PBIAS should be ±25% for the model performance to be considered satisfactory, while RSR values closer to 0 mean a more accurate model.

## 3 Results

### 3.1 MOHID-Land model

In natural regime flow, MOHID-Land's performance reached satisfactory to good results at Sar, Ulla, Arnego-Ulla and Deza
hydrometric stations (Table 4) as shown in Oliveira et al. (2020). The $R^2$ values ranged from 0.56 to 0.75 and 0.76 to 0.85 in the calibration (01/01/2009-31/12/2012) and validation (01/01/2013-31/12/2017) periods, respectively. The RSR showed values lower than 0.67 for all stations in both periods, while the NSE presented values from 0.55 to 0.72 in the calibration period and from 0.72 to 0.84 in the validation period. Finally, the PBIAS presented a slight overestimation of river flow in Sar hydrometric station (calibration: 0.18%; validation: 16.09%) while in the other three stations the model was
underestimating the river flow, with PBIAS values ranging from -12.29% to -8.96% and from -18.54% to -4.35% in calibration and validation periods, respectively.

**Table 4 Statistical indicators resulting from the comparison of the natural regime flow estimated by MOHID-Land with the observed streamflow values in 6 hydrometric stations (Cal. – calibration, Val. – validation, adapted from: Oliveira et al., 2020).**

Figure 7 compares the observed streamflow (black line) with the respective MOHID-Land simulations without considering
the influence of reservoirs (blue line) at Ulla-Touro and Ulla-Teo. Since these hydrometric stations have their natural regime



flow altered by the operation of the set of reservoirs in the watershed, the performance of the hydrological model without reservoirs showed a significative decrease, as expected (Table 4).

**Figure 7 Comparison of modelled and observed streamflow in hydrometric stations (a) Ulla-Touro and (b) Ulla-Teo with and without considering the existence of reservoirs.**

**3.2 CLSTM model**

To better evaluate the performance of CLSTM neural network model, the four statistical indicators were calculated for the set of 100 models trained with the same training dataset. Table 5 presents a summary of the results obtained.

**Table 5 Average, minimum, maximum and standard deviation values of the four statistical parameters estimated for the set of 100 models ran.**

The behavior of the developed CLSTM model was extremely regular, with an $R^2$ always above 0.89, and the NSE higher than 0.86. The worst PBIAS was -15.74%, and the maximum value of RSR was 0.37. More specifically, the trained model elected to represent the outflow estimation of Portodemouros reservoir obtained a NSE of 0.90, a $R^2$ of 0.91, a PBIAS of -2.61%, and a RSR of 0.31. Figure 8 shows the comparison between the modelled and the observed values for Portodemouros outflow.

**Figure 8 Comparison between modelled and observed Portodemouros outflow.**

The CLSTM predicted the outflow of Portodemouros reservoir very accurately. However, when the observed values showed accentuated differences in a short period of time, such as two consecutive days, the model demonstrated some difficulty in reproducing that behavior, being able to reproduce the increase-decrease behavior at the right instant but unable to reach correct values. This is the case of the outflow predictions for May and June of 2016 (Figure 8).

**3.3 Coupled system**

In the coupled system (MOHID-Land+CLSTM), Portodemouros outflow was estimated with the CLSTM model considering the level and inflow estimated by MOHID-Land model. Then, the outflow predicted by the CLSTM model was imposed in MOHID-Land. Therefore, the inflow and outflow of the reservoir as well as the two hydrometric stations influenced by the presence of the reservoirs were the target of the validation of the coupled system.

Figure 9a compares the observed and modelled inflow in Portodemouros reservoir, while Figure 9b shows the same comparison for the outflow. The observed (black line) and modelled (red line) streamflow comparison for Ulla-Tour and Ulla-Teo stations is presented in Figure 7a and Figure 7b, respectively. The four statistical indicators used to evaluate the model's performance were also calculated for the inflow, outflow, and the streamflow in Ulla-Touro and Ulla-Teo stations and are presented in Table 6.

**Figure 9 Comparison between the modelled and observed (a) inflow and (b) outflow in Portodemouros reservoir.**

**Table 6 Statistical parameters for inflow, outflow, and streamflow in Ulla-Touro and Ulla-Teo stations (Cal. – calibration, Val. – validation). The values between brackets represent the percentage of change of the statistical parameter to the corresponding value in the simulation without reservoirs.**



Inflows estimates in Portodemouros reservoir were in accordance with Oliveira et al. (2020). For the outflow values
estimated with the CLSTM model considering the original dataset, the performance of the coupled system slightly decreased
when compared with the previous indicators, with $R^2$ of 0.66, NSE of 0.55, RSR of 0.67, and PBIAS of -25% for the
validation period. The coupled system further showed a good performance when simulating streamflow in the two
hydrometric stations (Ulla-Touro and Ulla-Teo), which the regime flow is altered by the presence of the reservoirs.
Considering both hydrometric stations, the $R^2$ improved about 30% compared with the results without reservoir, reaching a
minimum of 0.70. The RSR indicator also demonstrated a better performance with values fitting the range from 0.39 to 0.63
and revealing an average improvement of about 30%. The higher impact was observed for the NSE indicator, which
increased about 253% with the values laying in the range from 0.61 to 0.85. Finally, the PBIAS showed an average decrease
of about 4%.

Despite the good results obtained for the streamflow downstream reservoirs, it is important to denote that the reservoir's
level estimated by MOHID-Land model did not reach the minimum requirements to be classified as satisfactory (calibration:
NSE=−2.44, $R^2$=0.01, PBIAS=−3.16%, RSR=1.85; validation: NSE=0.00, $R^2$=0.09, PBIAS=−0.67%, RSR=1.00). The
coupled system overestimated Portodemouros level most of the time, with exception for the period between January 2013
and the middle of 2016, when the observed and modelled values were more similar (Figure 10).

**Figure 10 Comparison between modelled and observed level in Portoudemouros reservoir.**

It could be expected that this issue would affect streamflow estimation downstream the reservoir since the outflow estimated
by CLSTM model considered the level values estimated by MOHID-Land. However, as demonstrated before, this issue did
not significantly impact downstream results.

**3.4 Impact of reservoirs 'operation on streamflow**

As referred before, the reservoirs have an impact on the natural regime flow downstream those infrastructures. The impact in
Ulla River watershed was here assessed by comparing the simulations under natural flow regime with the simulation of the
coupled system. For this, the streamflow was evaluated in three locations along the river network, namely, at the Ulla-Touro
and Ulla-Teo stations and at the outlet of the watershed. Table 7 shows the minimum, maximum, average, and 2nd, 3rd and 4th
quartiles values of the streamflow timeseries obtained for those locations considering the scenarios with (Res.) and without
(No res.) reservoirs.

**Table 7 Alterations on streamflow downstream reservoirs considering the simulations without and with those infrastructures (No
res. - without reservoirs; Res - with reservoirs).**

The most significative differences in streamflow occurred at the Ulla-Touro station, located immediately downstream the
reservoirs and more influenced by reservoirs' operations. The main differences between the two scenarios were observed in
the smallest values, namely, the minimum and the 2nd quartile. In both cases, the streamflow showed an increase when the
reservoirs were considered in the simulation, with the minimum streamflow increasing 105% in the outlet, 127% in Ulla-
Teo, and 356% in Ulla-Touro, and the 2nd quartile increasing 16%, 17% and 28% in the outlet, Ulla-Teo and Ulla-Touro,



respectively. On the opposite, the main decreases were observed in the maximum and 4th quartile for all the evaluated points. However, the decreases of the highest values were not so significant as the differences observed for the smallest values, with the maximum values decreasing 10% in the outlet, 6% in Ulla-Teo and 18% in Ulla-Touro and the 4th quartile presenting differences of -3%, -4% and -6% in the outlet, Ulla-Teo and Ulla-Touro, respectively.

The distribution of streamflow along the year (Figure 11) showed a decrease in the average streamflow between October and December (wet season) when considering the reservoirs. Between January and March, also in the wet season, the streamflow only showed slight differences when considering or not the reservoirs. Finally, the dry season was totally characterized by an increase in the streamflow for the simulations with reservoirs, with the main differences found between July and September. For the same reasons presented before, Ulla-Touro station was the point where the main differences were observed.

**Figure 11 Average monthly streamflow in Ulla-Touro and Ulla-Teo stations and in the outlet for the two simulated scenarios, i.e., without and with reservoirs.**

The behaviour presented in Figure 11 is the expected result when considering reservoirs' operations since this type of infrastructure are commonly used to store water during the wet season, causing a decrease of downstream streamflow. On the other hand, it is expected that average streamflow increases during dry seasons due to the constant necessity of energy production throughout the year and the imposition of ecological flows downstream reservoirs to maintain the health of the ecosystems.

## 4 Discussion

The results of the presented study show that the direct incorporation of reservoirs' operation in hydrologic modelling has a significative impact on the results of the modelled system, as already referred by Bellin et al. (2016). The development of the CLSTM model to predict Portodemouros outflow, which was after imposed in the hydrological model, needed to guarantee that the model estimation was good enough to avoid an error propagation. The elected CLSTM model reached a performance where the NSE was 0.90, the $R^2$ was 0.91, and the PBIAS and RSR were -2.61% and 0.31, respectively, considering a test dataset. Similar results were obtained by Yang et al. (2019), who estimated the daily outflow of three multipurpose reservoirs located in Thailand, considering three different types of RNN models, namely, a non-linear autoregressive model with exogenous input (NAXR), a long short-term memory (LSTM), and a genetic algorithm based on NAXR (GA-NAXR). The authors considered as forcing variables the inflow estimated by a hydrological model in the previous two days and the following two days together with the reservoir storage volume in the previous day. They obtained an average Pearson correlation coefficient of 0.91, an average NSE of 0.81, and an average PBIAS of -0.71% with the NARX model and considering the three modelled reservoirs. The LSTM and GA-NARX models reached an average Pearson correlation coefficient of 0.88 and 0.94, respectively, an average NSE of 0.72 and 0.88 and an average PBIAS of 0.22% and -0.24%, with the GA-NARX demonstrating the best performance. Hughes et al. (2021) demonstrated the ability of a modified version of the SHETRAN model to predict the outflow of Crummock Water Lake, located in the Upper Cocker catchment, in United



Kingdom. By including a dynamic weir module in the original SHETRAN model, the authors deducted the behavior of
sluices by comparing the outflow values of a static and a dynamic weir models. The developed approach reached an NSE of
0.82, a value similar to the ones obtained in the present study, but its application to other case studies presents several
limitations. First, it can be very laborious since it was based on a generic framework that included 12 steps. Second, the
implementation of that framework implied a deep knowledge about the geometry of control structures and the details of
operating procedures, with the authors referring that the broad conceptual understanding of sluice operations needed for the
implementation was obtained through site visits and operator interviews.

On the other hand, the estimation of reservoirs' outflow using neural network models, such as the CLSTM model used here
can also contain several limitations. With the choice of the forcing variables being pointed out by several authors as crucial
for a successful model (ASCE, 1996; Maier et al., 2010; Dolling and Varas, 2002; Wu et al., 2014; Juan et al., 2017), the
consideration of other forcing variables should be evaluated. Also, the structure of this type of model, that includes the
number of hidden layers, the number of nodes, the kernel size, the activation functions, and other characteristics, is usually
optimized by a trial-and-error procedure. However, the number of options that can be adopted for each of those structural
characteristics and their combination makes the search space too wide to evaluate all the possible solutions. Thus, the manual
approach adopted here to define the model's structure can be restrictive to the searching of the best solution since a small
number of possible solutions were tested when considering the entire search space. It is then clear that the optimization of
the structure of CLSTM model can improve the results. As suggested by Oliveira et al. (2023), this task can be done using
tools that implement different algorithms to efficiently search for the best solution contained in a search space.

Considering the coupled system, the results showed a very clear and interesting improvement when compared with the
implementation without reservoirs, with all the statistical indicators demonstrating a better performance in the coupled
system for the two hydrometric stations influenced by reservoirs' operations. Although the coupled system has demonstrated
a very good performance it is important to refer that besides the limitations already pointed to the CLSTM model, the
coupled system has its own limitations. Firstly, when CLSTM is incorporated into the system it will use an estimated inflow,
in opposition with the observed values used to train the model. Thus, when the inflow value is not correctly estimated by the
hydrologic model it will negatively influence the estimation of the outflow by the CLSTM model, leading to an exacerbation
of the error downstream this point. Also, the level used by the CLSTM model to force the outflow estimation is simulated by
a mass balance performed by the hydrologic model. However, MOHID-Land does not yet incorporate the reservoir's loss by
evaporation and infiltration, which can lead to an overestimation of the reservoir's level as observed in Figure 10. As in the
case of the inflow, if the level value that feeds the CLSTM model is far from the correct value, the estimated outflow will
also be inaccurate and may lead to an increased error in downstream areas.

Nevertheless, the results of this study agree with other studies. For instance, Yun et al. (2020) modified the original VIC
model to contemplate the reservoirs 'operations in the Lancang-Mekong River basin, in Asia, and compared the performance
of the model with observed data in five hydrometric stations. Considering the calibration and validation periods, the author
obtained NSE values ranging from 0.61 and 0.75 and a model bias that varied between -3% and 4% for daily streamflow.



Following a similar approach, Dang et al. (2020) modified the VIC model to integrate reservoirs' operation into hydrological simulations. 118 solutions of the model with reservoirs and 109 solutions without reservoirs were run and automatically 480 calibrated considering the upper Mekong River basin as case study. That set of models obtained NSE values from 0.68 to 0.79, and a transformed root mean square error from 8.10 to 16.69, with the statistics of both solutions evenly distributed in those ranges. It is important to denote that the authors referred that, in the case of the implementations without reservoirs, the model reached such good performance probably because the model parameterization helped to compensate the structural error of the non-consideration of reservoirs. However, in both modified versions of the VIC model, reservoirs' operations 485 were imposed by the authors through the definition of several operation rules that implied the knowledge of reservoirs' characteristics that sometimes are not easily available, such as the normal storage, the flood-limited storage, the environmentally friendly streamflow, the maximum safe streamflow for the downstream area, the capacity of the turbines, the target storage, and others. This fact can limit the application of both methodologies in areas with limited information.

Dong et al. (2023) adopted a similar approach to the one presented in this study, using two data-driven models to reproduce 490 reservoirs behavior in terms of outflow, when data was available, and coupled them with a high-resolution model. For the reservoirs with no data, a calibration-free conceptual reservoir operation scheme was designed. Considering the Upper Yangtze River Basin, China as a case study, 10 reservoirs were considered, with 4 being simulated with the data-driven models and 6 being simulated with the conceptual scheme. The authors simulated the outflow and the storage of the reservoirs using a XGBoost model and an ANN model, with the first demonstrating the best performance for both properties. 495 Considering the test period, XGBoost obtained NSE values higher than 0.85 for the outflow simulation and higher than 0.88 for the storage simulation, while the same indicator was higher than 0.80 and 0.83 for the outflow and storage simulations, respectively, when the ANN was considered. Taking into account the set of hydrometric stations analyzed, the NSE values were higher than 0.65.

Finally, the reservoir's downstream effects on streamflow values found in this study were also in accordance with Yun et al. 500 (2020) and Dong et al. (2023). Both authors concluded that the presence of the reservoirs decreased the average streamflow during the wet season and increased in the dry season, with a higher increase during the dry season than the decrease in the wet season. In Ulla River basin, the annual average streamflow did not verify any changes; however, the differences in wet and dry seasons were also observed (Figure 11). During the wet season (Oct-Mar), the streamflow suffered a decrease of about 5%, 3% and 2% in Ulla-Touro, Ulla-Teo and in the outlet of the watershed, respectively. For the dry season (Apr-Sep), 505 increases of approximately 18%, 9% and 8% were estimated for those same points. At the same time, the maximum streamflow and the 4th quartile verified a decrease when the presence of the reservoirs was considered. The maximum streamflow decreased a maximum of 18% (from 319 $m^3$ $s^{-1}$ to 261 $m^3$ $s^{-1}$) in Ulla-Touro station and a minimum of 6% (from 462 $m^3$ $s^{-1}$ to 433 $m^3$ $s^{-1}$), while the 4th quartile presented decreases between 6% (from 44 $m^3$ $s^{-1}$ to 41 $m^3$ $s^{-1}$) and 3% (from 99 $m^3$ $s^{-1}$ to 96 $m^3$ $s^{-1}$) at Ulla-Touro and at the outlet, respectively. The capacity of decreasing and control flow peaks is of 510 extreme importance in Ulla River basin, since the downstream area is exposed to high flood risk exacerbated by the combination of intense rainfall events and the influence of high tides (Augas de Galicia, 2019).



**5 Conclusion**

The approach presented and discussed in this work comprehended the direct integration of reservoirs operation into a hydrologic model. A CLSTM data-driven model was developed to estimate the reservoirs outflow, which values were then imposed in the MOHID-Land model. The case study focused on the Ulla River basin, which was the target of a previous work where MOHID-Land was implemented, calibrated, and validated for natural regime flow. In this watershed, a set of three reservoirs are present, with the one more upstream having the higher storing capacity while the following two work as run-of-the-river dams. The operation of run-of-the-river dams was simulated with an operation curve that relates the level, the inflow and the outflow of the reservoirs, and the outflow of the high-capacity reservoir was estimated using the CLSTM model. The target of this work was to analyze how streamflow simulations improved in the areas where the natural regime flow was modified by reservoirs' operations using the proposed coupled system. The main conclusions were:

1. The CLSTM model selected to represent Portodemouros' outflow showed a very good performance, with NSE, $R^2$ and RSR values of 0.90, 0.91, and 0.31, respectively. The PBIAS was -2.61% indicating a very slight underestimation of the reservoir outflow.

2. The implementation of the coupled system demonstrated a significative improvement of streamflow estimations in areas downstream reservoirs, with the NSE increasing from a minimum of -0.09 without reservoirs to a minimum of 0.61 with reservoirs. Also, the $R^2$ demonstrated the same improvement, increasing from a minimum of 0.46 to 0.70 without and with reservoirs, respectively.

3. The MOHID-Land model failed to reproduce the level of the high-capacity reservoir, in part because the model does not include evaporation losses. However, the lack of accuracy did not have a significative impact on the performance of the coupled system.

4. According to the validation performed downstream reservoirs, the basic operation curves selected to simulate the function of the two run-of-the-river dams in the study domain seemed adequate.

5. For the modelled 10-year period, the impacts downstream reservoirs were in line with other studies, with the maximum streamflow (wet season) values experiencing a decrease and the minimum values (dry season) suffering an increase. However, the average streamflow did not show any increase or decrease tendency.

Besides the excellent results obtained in this study, future applications of the methodology should be tested and evaluated to understand its applicability to different scenarios. One of the doubts that remains is if the CLSTM model has the capacity to reproduce the behavior of a reservoir where water is used for irrigation, which is characterized by punctual discharges in time, instead of an almost continuous discharge as in Portodemouros. Also, the capability of the trained CLSTM model in reproducing outflow values of other reservoirs with similar purposes should be addressed.



**Code availability:** MOHID-Land is available on the GitHub repository https://github.com/Mohid-Water-Modelling-System/Mohid. The CLSTM model and the scripts to run the coupled system are available on 545 https://github.com/anaioliveira/NNandMOHID.

**Author contributions:** A.R.O. and L.P. were responsible for the conceptualization. A.R.O. performed the formal analysis, developed and implemented the methodology and the software, and was responsible for results analysis and the visualization. T.B.R. and R.N. were responsible for the funding acquisition. The original draft was written by A.R.O. and 550 revised by T.B.R., L.P. and R.N.

**Competing interests:** The authors declare that they have no conflict of interest.

**Funding:** This research was supported by FCT/MCTES (PIDDAC) through project LARSyS–FCT pluriannual funding 555 2020–2023 (UIDP/50009/2020), and by Project FEMME (PCIF/MPG/0019/2017). T. B. Ramos was supported by a CEEC-FCT contract (CEECIND/01152/2017).

**Acknowledgments:** We would also like to thank Augas de Galicia for reservoirs' data availability. Without their support this work would be impossible feasible.

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

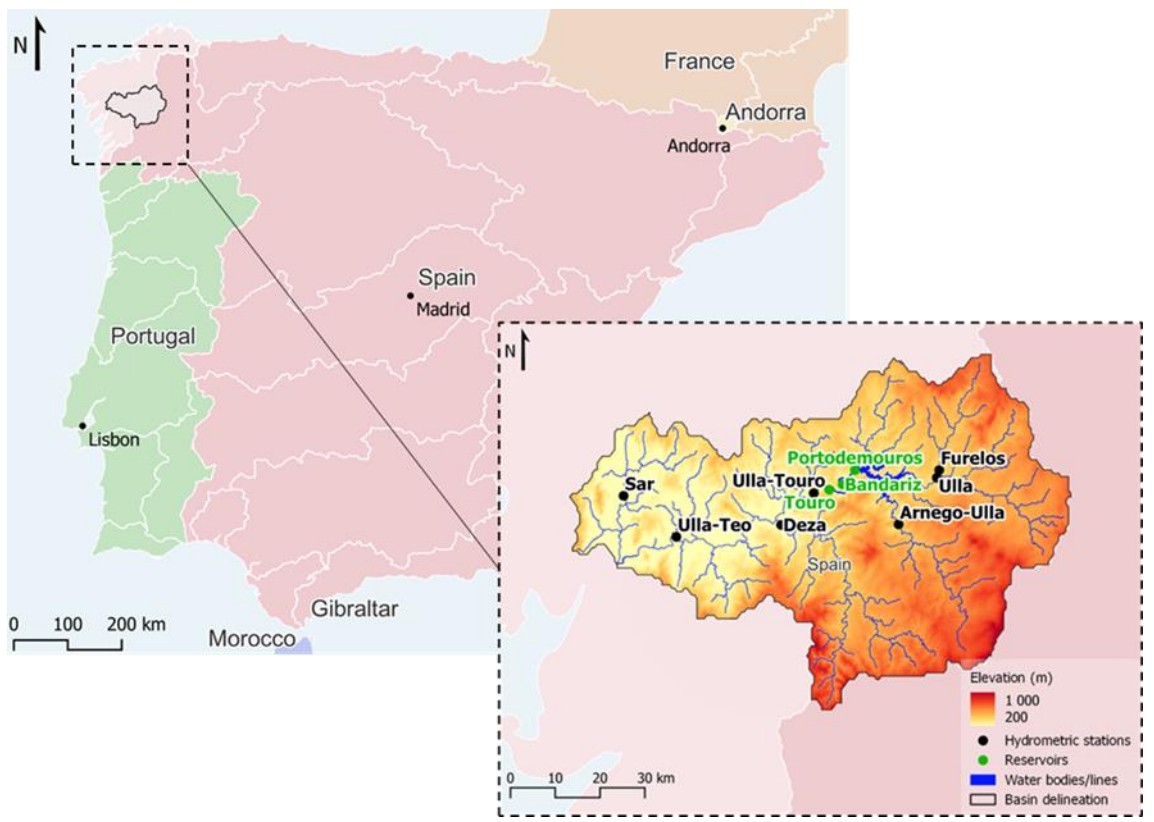

**Figure 1 Ulla River watershed location, digital terrain model, and identification of hydrometric stations and reservoirs.**



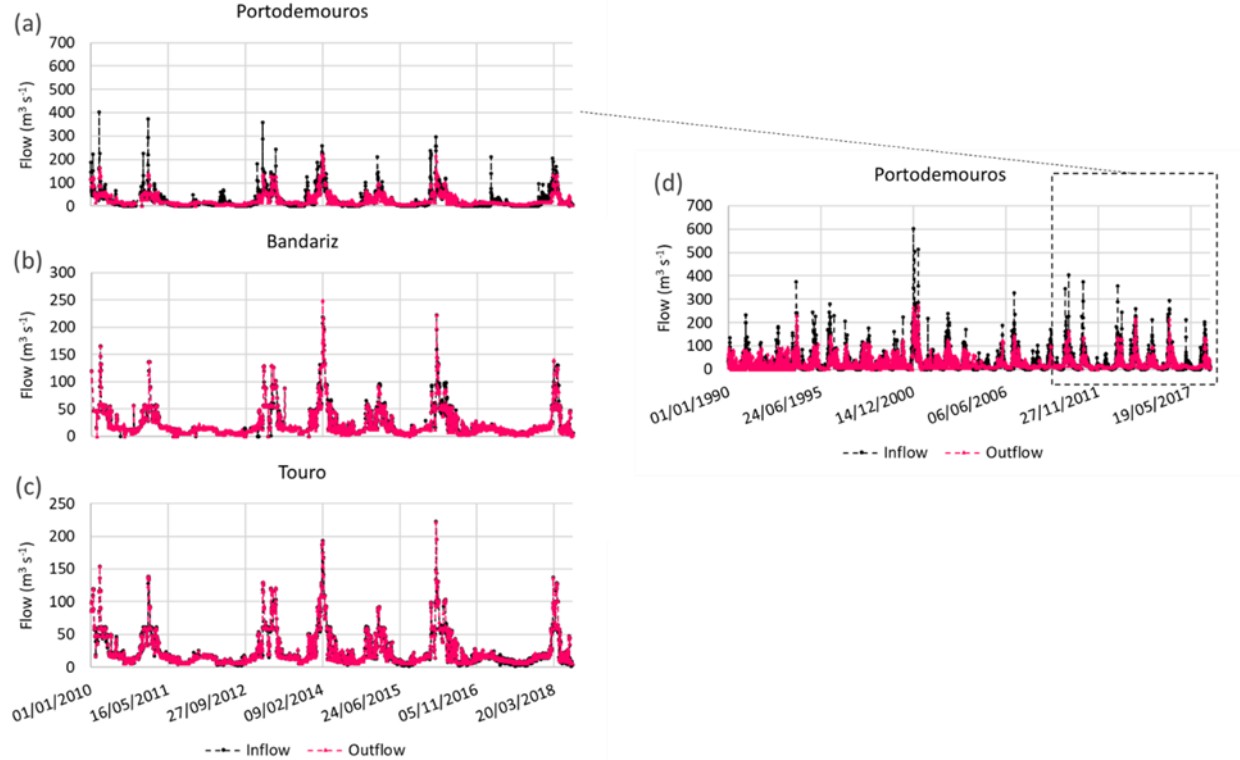

**Figure 2 Comparison of inflow and outflow in (a) Portodemouros, (b) Touro, and (c) Bandariz reservoirs for the period 2010-2018, and in (d) Portodemouros reservoir for the period 1990-2018.**


| Drained area | Top width | Depth |
|---|---|---|
| (km²) | (m) | (m) |
| 37.85 | 12.71 | 2.0 |
| 62.65 | 16.45 | 2.0 |
| 84.49 | 19.16 | 2.0 |
| 123.35 | 23.24 | 3.0 |
| 161.90 | 26.71 | 3.0 |
| 195.10 | 29.38 | 3.0 |
| 312.45 | 37.36 | 3.0 |
| 503.12 | 46.95 | 4.0 |
| 1164.36 | 73.16 | 4.0 |
| 2246.34 | 102.33 | 4.0 |
| 2785.08 | 114.21 | 4.0 |

**Table 1 Cross-sections dimensions.**



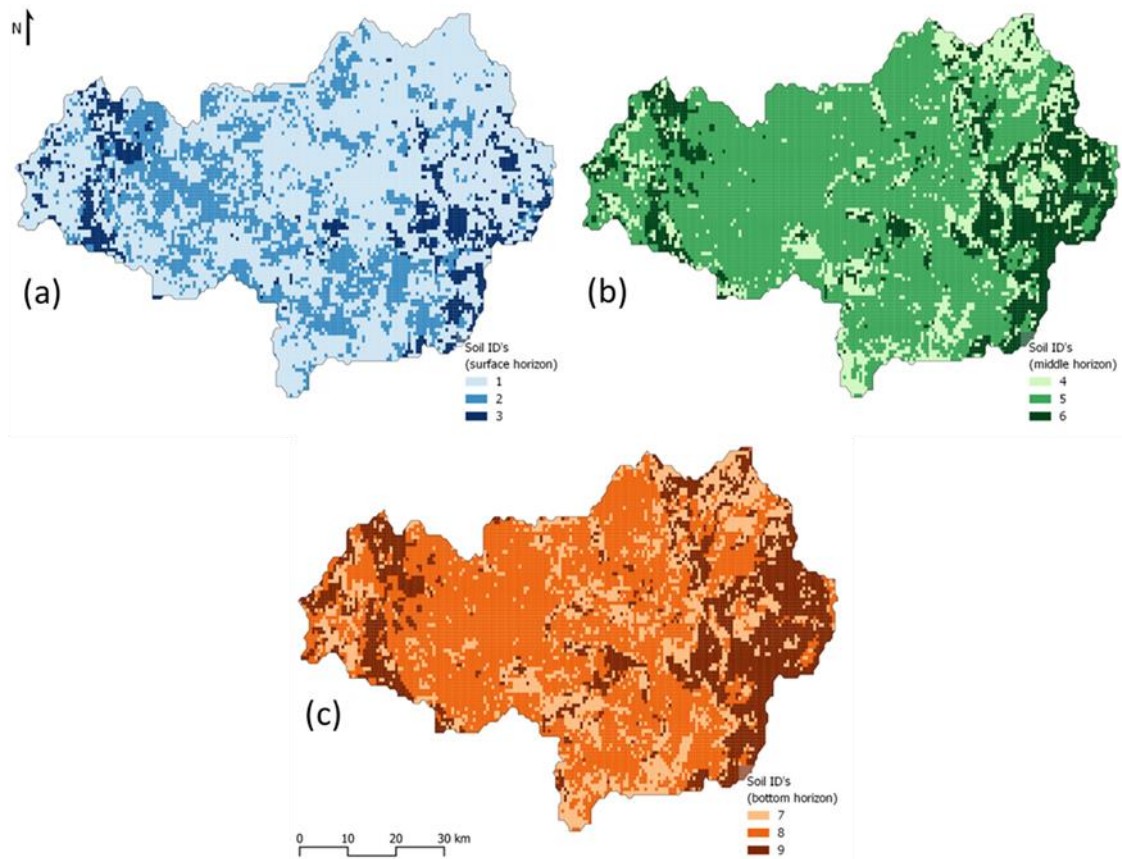

**Figure 3 Soil IDs for each horizon: (a) surface; (b) middle; and (c) bottom horizons.**


| ID | $\theta_s$ <br> (m³ m⁻³) | $\theta_r$ <br> (m³ m⁻³) | $\eta$ | $K_{sat,ver}$ <br> (m³ s⁻¹) | $\alpha$ <br> (m⁻¹) | l |
|----|------|------|-------|-----------|-------|------|
| 1 | 0.491 | 0.0 | 1.913 | $1.64 \times 10^{-5}$ | 3.47 | -4.3 |
| 2 | 0.465 | 0.0 | 1.116 | $2.26 \times 10^{-4}$ | 12.84 | -5.0 |
| 3 | 0.409 | 0.0 | 1.134 | $5.05 \times 10^{-5}$ | 7.00 | -5.0 |
| 4 | 0.433 | 0.0 | 1.170 | $9.93 \times 10^{-6}$ | 3.36 | -5.0 |
| 5 | 0.413 | 0.0 | 1.119 | $1.43 \times 10^{-5}$ | 2.27 | -5.0 |
| 6 | 0.384 | 0.0 | 1.121 | $4.29 \times 10^{-5}$ | 7.17 | -5.0 |
| 7 | 0.432 | 0.0 | 1.170 | $9.93 \times 10^{-6}$ | 3.36 | -5.0 |
| 8 | 0.413 | 0.0 | 1.119 | $1.43 \times 10^{-5}$ | 2.27 | -5.0 |
| 9 | 0.384 | 0.0 | 1.121 | $4.29 \times 10^{-5}$ | 7.17 | -5.0 |

**Table 2 Soil hydraulic properties by soil ID: $\theta_s$ – saturated water content; $\theta_r$ – residual water content; $\eta$ and $\alpha$ – empirical shape parameters; $K_{sat,ver}$ – vertical saturated hydraulic conductivity; and l – pore connectivity/tortuosity parameter.**

| | Portodemouros | Bandariz | Touro |
|---|---|---|---|





| Node location | 1476 | 1383 | 1247 |
|---|---|---|---|
| Coordinates | 42°51'21.6"N 8°11'19.8"W | 42°50'09.6"N 8°12'31.8"W | 42°49'51.6"N 8°14'19.8"W |
| Minimum volume (hm$^3$) | 54.5 | 0.33 | 0.015 |
| Maximum volume (hm$^3$) | 297 | 2.74 | 6.83 |
| Minimum outflow (m$^3$ s$^{-1}$) | 10 | 10 | 10 |

**Table 3 Implemented characteristics for Portodemouros, Bandariz and Touro reservoirs.**

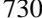

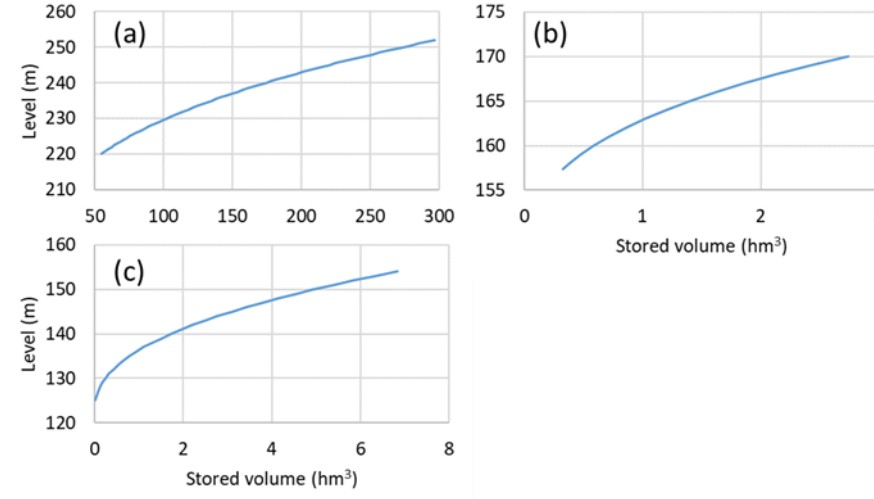

**Figure 4 Level/stored volume curves for (a) Portodemouros, (b) Bandariz, and (c) Touro reservoirs.**

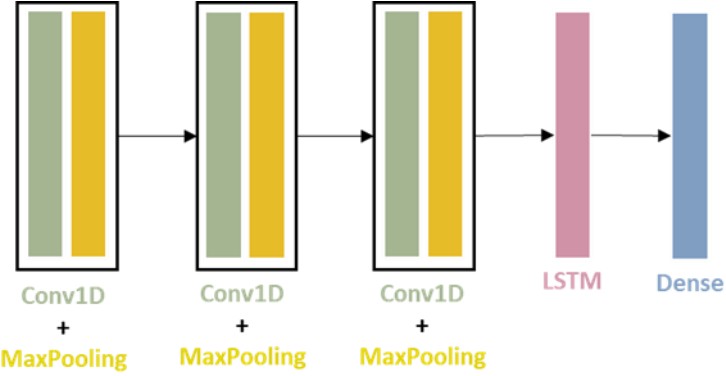

**Figure 5 CLSTM structure.**



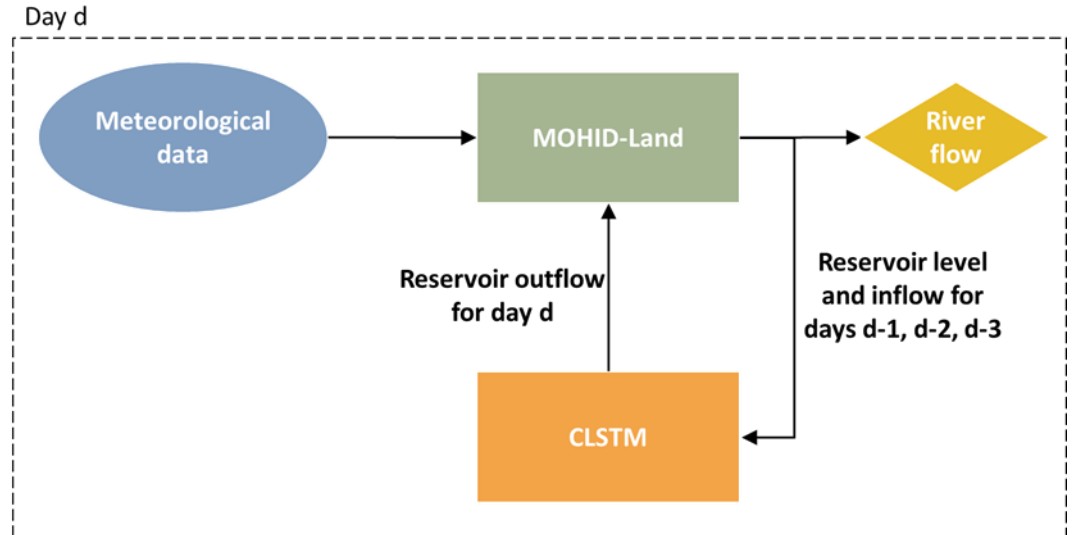

**Figure 6 Operationality scheme for the modelling process.**

| Station | R² (-) | | NSE (-) | | RSR (-) | | PBIAS (%) | |
|---|---|---|---|---|---|---|---|---|
| | Cal. | Val. | Cal. | Val. | Cal. | Val. | Cal. | Val. |
| Sar | 0.75 | 0.83 | 0.72 | 0.81 | 0.53 | 0.44 | 0.18 | 16.09 |
| Ulla | 0.56 | 0.76 | 0.55 | 0.72 | 0.67 | 0.53 | -11.24 | -18.54 |
| Arnego-Ulla | 0.70 | 0.78 | 0.69 | 0.76 | 0.55 | 0.49 | -12.29 | -16.82 |
| Deza | 0.74 | 0.85 | 0.72 | 0.84 | 0.53 | 0.40 | -8.96 | -4.35 |
| Ulla-Touro | 0.46 | 0.52 | -0.09 | 0.24 | 1.04 | 0.87 | -19.06 | -19.12 |
| Ulla-Teo | 0.77 | 0.79 | 0.71 | 0.73 | 0.54 | 0.52 | -16.68 | -14.36 |

**Table 4 Statistical indicators resulting from the comparison of the natural regime flow estimated by MOHID-Land with the observed streamflow values in 6 hydrometric stations (Cal. – calibration, Val. – validation, adapted from: Oliveira et al., 2020).**



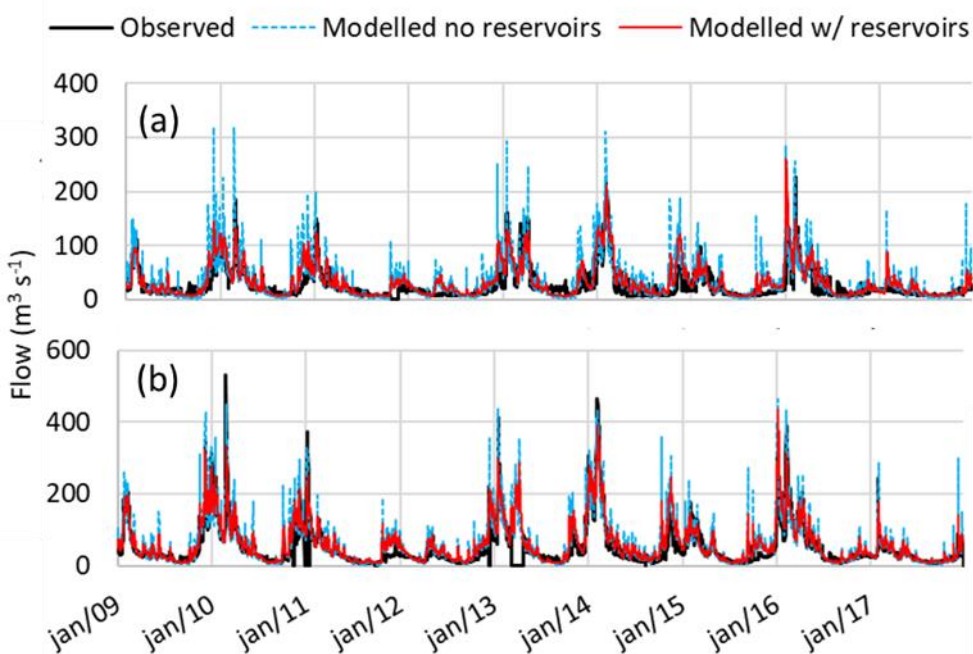

**Figure 7 Comparison of modelled and observed streamflow in hydrometric stations (a) Ulla-Touro and (b) Ulla-Teo with and without considering the existence of reservoirs.**

|  | R² (-) | NSE (-) | RSR (-) | PBIAS (%) |
|---|---|---|---|---|
| **Average** | 0.90 | 0.89 | 0.33 | -1.71 |
| **Minimum** | 0.89 | 0.86 | 0.31 | -15.74 |
| **Maximum** | 0.91 | 0.90 | 0.37 | 14.07 |
| **Standard deviation** | 0.00 | 0.01 | 0.01 | 6.26 |

**Table 5 Average, minimum, maximum and standard deviation values of the four statistical parameters estimated for the set of 100 models ran.**





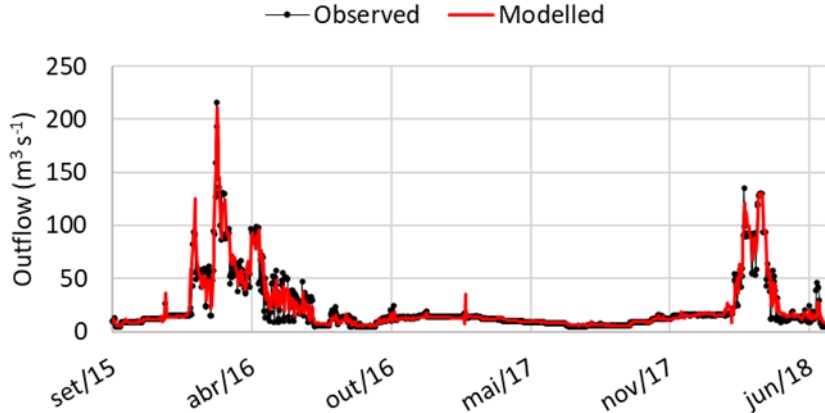

**Figure 8 Comparison between modelled and observed Portodemouros outflow.**

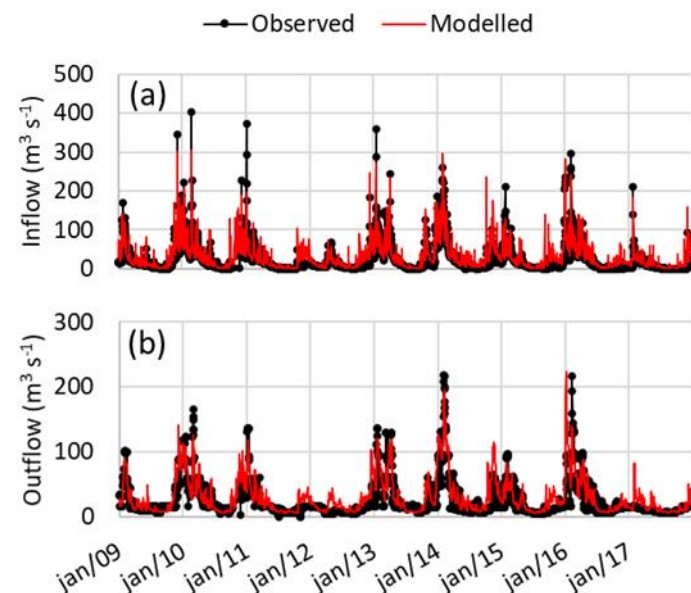

**Figure 9 Comparison between the modelled and observed (a) inflow and (b) outflow in Portodemouros reservoir.**

| Station | R² (-) | | NSE (-) | | RSR (-) | | PBIAS (%) | |
|---|---|---|---|---|---|---|---|---|
| | Cal. | Val. | Cal. | Val. | Cal. | Val. | Cal. | Val. |
| **Inflow** | 0.79 | 0.81 | 0.76 | 0.77 | 0.49 | 0.48 | -23.68 | -28.38 |
| **Outflow** | 0.71 | 0.66 | 0.64 | 0.55 | 0.60 | 0.67 | -19.53 | -25.35 |
| **Ulla-Touro** | 0.74 (+61%) | 0.70 (+35%) | 0.65 (+822%) | 0.61 (+154%) | 0.59 (-43%) | 0.63 (-28%) | -17.20 (-10%) | -19.58 (+2%) |
| **Ulla-Teo** | 0.87 | 0.86 | 0.85 | 0.83 | 0.39 | 0.41 | -15.48 | -14.68 |



| (+13%) | (+9%) | (+20%) | (+14%) | (-28%) | (-21%) | (-7%) | (+2%) |

**Table 6 Statistical parameters for inflow, outflow, and streamflow in Ulla-Touro and Ulla-Teo stations (Cal. – calibration, Val. – validation). The values between brackets represent the percentage of change of the statistical parameter to the corresponding value in the simulation without reservoirs.**

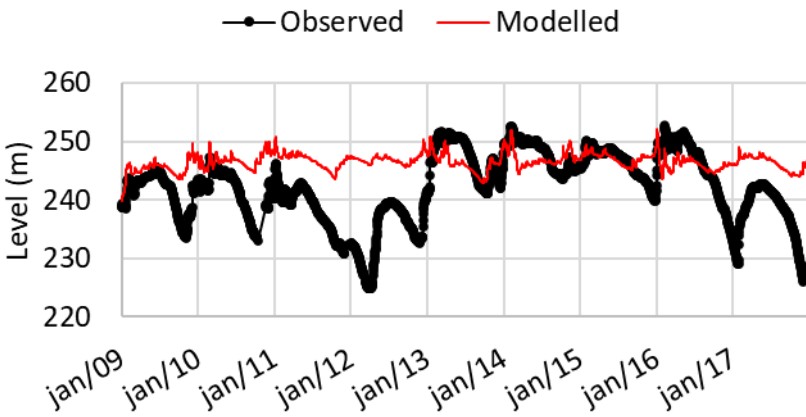


**Figure 10 Comparison between modelled and observed level in Portoudemouros reservoir.**

| Statistical indicator | Ulla-Touro | | Ulla-Teo | | Outlet | |
|---|---|---|---|---|---|---|
| | **No res.** | **Res.** | **No res.** | **Res.** | **No res.** | **Res.** |
| **Minimum (m³ s⁻¹)** | 1.4 | 6.2 (+356%) | 3.6 | 8.2 (+127%) | 4.2 | 8.7 (+105%) |
| **Maximum (m³ s⁻¹)** | 319.1 | 260.8 (-18%) | 462.2 | 432.8 (-6%) | 569.3 | 511.9 (-10%) |
| **Average (m³ s⁻¹)** | 33.4 | 33.2 (-1%) | 62.1 | 62.1 (0%) | 74.1 | 73.9 (0%) |
| **2ⁿᵈ quartile (m³ s⁻¹)** | 8.5 | 10.9 (+28%) | 17.2 | 20.1 (+17%) | 20.2 | 23.4 (+16%) |
| **3ʳᵈ quartile (m³ s⁻¹)** | 21.5 | 22.0 (+3%) | 42.0 | 42.8 (+2%) | 49.7 | 50.2 (+1%) |
| **4ᵗʰ quartile (m³ s⁻¹)** | 43.5 | 40.9 (-6%) | 83.3 | 79.8 (-4%) | 99.2 | 96.0 (-3%) |

**Table 7 Alterations on streamflow downstream reservoirs considering the simulations without and with those infrastructures (No res. - without reservoirs; Res - with reservoirs).**






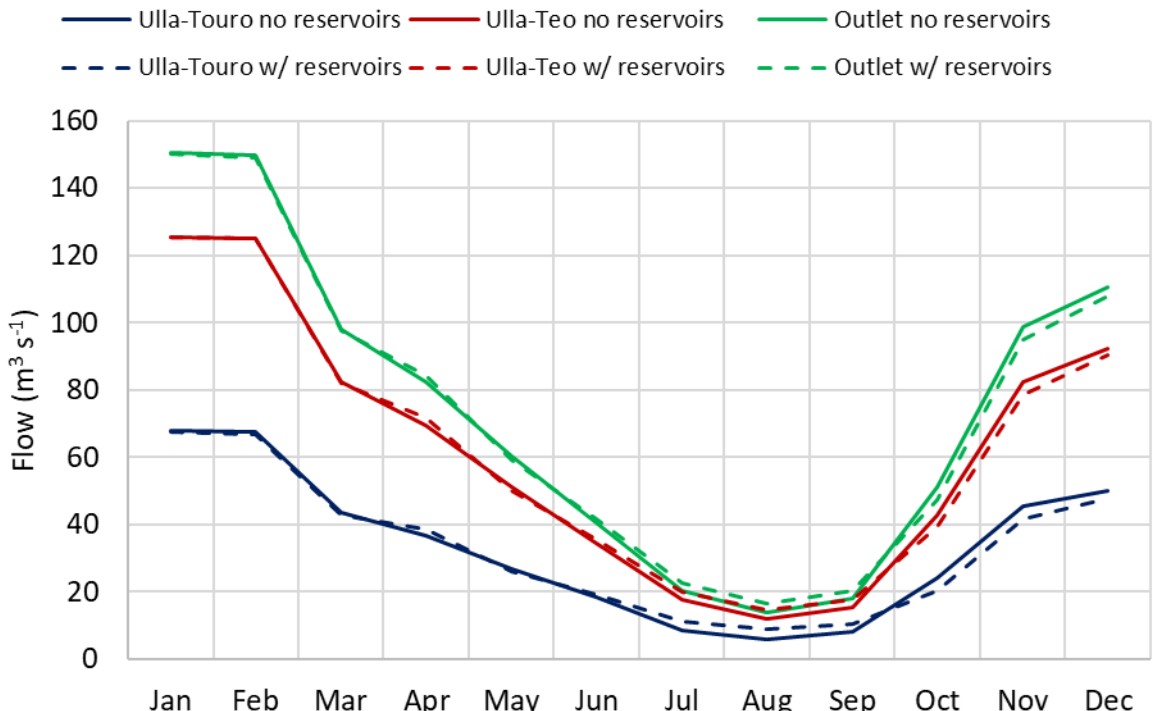

**Figure 11 Average monthly streamflow in Ulla-Touro and Ulla-Teo stations and in the outlet for the two simulated scenarios, i.e., without and with reservoirs.**