# Peer review of "Direct integration of reservoirs' operations in a hydrological model for streamflow estimation: coupling a CLSTM model with MOHID-Land"

_EGUsphere, 2023_

## Referee Comment (RC1)

**Referee comment**

**General comments**

The manuscript presents an interesting and relevant work addressing scientific questions within the scope of HESS. The authors present a successful newly implemented coupled system with the main objective of estimating the outflow from a reservoir in Spain. The methodology developed and applied merges the physical understanding of a physical-based model and the capabilities of artificial neural networks to learn from non-linear processes. Therefore, applying a coupled system highlights the novelty of the work, which is not simply another application of a hydrological model or an artificial neural network to estimate streamflow in a given catchment. In this sense, the authors gave proper credit to related work and clearly indicated their original contribution. The manuscript is well-structured, clear and concise. It is well-written and easy to understand. The authors present a good introduction of state of the art regarding the subject and a well-written description of the study area and the materials ad methods applied. The studies used for introducing the subject and the discussion are relevant and recent. The results presentation and discussion gave a clear and sufficient overview of the results and their respective limitations. The conclusions present an interesting closing of the presented workflow concisely and straightforwardly, but not forgetting to present the future perspectives from this work. Finally, the authors present the workflow code in a public and open repository, contributing significantly to open science and making it possible the work reproducibility. Therefore, the paper deserves to be published in HESS after some corrections and adaptations regarding the guality of the writing and figures.

**Specific comments**

In this section, I present some specific comments to be answered and implemented in the manuscript. No changes in the methodology are proposed, only suggestions for the writing, discussion and overall figures and tables quality.

- 1. The authors provided a complete public repository with the workflow. I recommend not going into too much detail within the text regarding the specific Python libraries and functions applied in this methodology.
- 2. Likewise, regarding the MOHID-Land model, since the model was already implemented in the referenced work of Oliveira et al. (2020) where it was fully described, the information regarding the implementation not already presented in this work should be given in the present details in Annex instead of in the main text. The authors can consider only a summary of the information instead of being in such details in the present work.
- 3. The authors conclude that the poor representation of reservoir levels reflects the non-inclusion of evaporative losses in the model (L529). I agree, but do the authors have concrete results corroborating this? Did you perform tests including it in your model? Would that be a possibility for further work? For this work, it is optional to make this inclusion. However, more evidence could improve your conclusions.
- 4. I would like to have a deeper discussion about why, even with the limitation in reproducing the reservoir levels, the model still performed well in reproducing the streamflow at the outlet. Is this due somehow to the concentration time of the watershed? A slight increase in the discussion will enrich your paper.
- 5. L361: What would cause this behavior by the model in your option? Could it be further discussed in the text?
- 6. L452: In this part of the discussion, the authors claim several limitations to using the CLSTM model. The text follows, describing several concerns, but are those the limitations intended by the authors to be mentioned here? I recommend rewriting this part text for clarification.
- L459-461: The authors claim that the optimization of the CLSTM could improve the results. Why
  was this not tested for this work? I would appreciate a further discussion of this matter in the
  manuscript text.

**Technical corrections**

In this section, I present some technical corrections and suggestions to be implemented in the text. This section is divided into an initial general part and later divided by each text section (e.g., Abstract, Introduction).

**General:**

Please refer to the guidelines template available at (https://www.hydrology-and-earth-system-sciences.net/submission.html#templates) when reviewing your manuscript.

**Symbols and variables:**

Consider a full check in all the symbols, such as "m", "Km" etc., to ensure they are correctly placed near their respective words. Consider using a standard number of decimals in nearby sentences, e.g., "11.6 and 2.5m", and not "11.6 and 2.55m".

**Equations and variables:**

For all equations, please make sure to cite them in the text before presenting them using the HESS guidelines for equations, i.e., using the label "Eq. (X)" or "Equation (X)" depending on the situation. It is a better practice to refer to your equation as:"...using Eq. (X):" instead of solely mention "...using the equation below:". Additionally, Make sure to have all variables in italics when mentioned in the full text, e.g.,  $x_{scaled}$  and not  $x_{scaled}$ .

**Figures and tables:**

Please consider improving the figures and tables' quality regarding font size, style and colors. The figures should have a similar style, and the authors can think of improving their quality following some suggestions in the further sections. Please ensure that all figures and tables located in the main text are mentioned in the text. Finally, please move some tables and figures to the annexes and reduce the final number of figures in the manuscript's main text by merging them when possible.

**Abstract**

L10: Re-word from "estimate" to "estimation".

L12: Please present the main objective of the manuscript more clearly. Beginning the presentation of your objective in the abstract with: "In this study, and already implemented..." does not capture the main objective of this study which is the development and application of a coupled system. Please start your objective presentation by making sure to present this information. Make sure to have your abstract following this pathway: Contextualization/Relevance – Objective – Methodology – Results – Conclusions.

L13: Please consider deleting the part: "..., calibrated, and validated solution of..."

L18: Re-word from "This coupled system was daily evaluated in two hydrometric stations..." to "This coupled system was evaluated on a daily basis using two hydrometric stations...".

L22: Consider changing "modified rivers" to "modified catchments".

**1** Introduction**

L24-28: Consider to summarize this list of activities and situations.

L31: Delete "mechanisms". It was already mentioned and it is redundant. Forcing is enough.

L37: Re-word from "makes it impeditive" to "poses a challenge".

L42: Delete the extra space after "faced".

L54-56: Consider using numbering to list those two tests, i.e., "(a) first the weir was simulated as static (with closed sluice) to identify the sluice 55 operating rules by comparing results with the known outflow time series; (b) second the weir model was run as non-static to implement the sluice operating rules deducted from the first approach."

L56: Consider start a new paragraph after the previous mentioned statements.

L76: I believe that if you keep using "the physical-based..." you should refer the official MOHID-Land reference, and not the previous work. However, if you re-word to "a physical based distributed Mohid-Land..." you may keep the reference to your previous work.

L80: Consider moving "However, the CLSTM model was first trained and tested using historical data." to L78, after you introduced the CLSTM model.

**2 Materials and methods**

**2.1 Description of the study area**

L90: Consider re-word from: "-0.75m" to "-1m". Is it not necessary to be so precise at this description.

L92: Insert a reference for the Köppeb-Geiger classification.

L102: Clarify what you mean by "set of reservoirs". The other two reservoirs? Or the whole set?

L104: Consider to use a synonym for "however" since it was already mentioned nearby.

**Figure 1:** Consider inserting the "Atlantic ocean" and "Mediterranean sea" labels on the figure. The palette choice for the countries is not colorblind friendly (consider alternative colors for green and red together). It is also a little difficult to identify the reservoirs in the figure. Consider altering the font sizes with their respective names, or moving these labels. Re-word "water bodies/lines" to "water courses". Re-word "Basin delineation" to "Ulla river watershed" for clarity. Check the legend color bar labels. Are those correct? Finally, it is missing the latitude and longitude or X and Y coordinates of the study area. They should be inserted in the map.

**Figure 2:** You have four subplots here. I would consider two changes: (1) subplots a-c should have the same Y-scale, e.g., 0-450 m3s-1. Subplot d can keep its scale since it refers to a different time-period.

**2.2 MOHID-Land description & 2.2.1 Reservoirs module**

As this work uses the already implemented model from Oliveira *et al.* (2020), I would consider a summarization of these sections, and if needed the addition of this full description in Annex. This description is interesting to understand the MOHID-Land implementation; however, it is too long for this section and for the scope of the present work. In Annex, you can additionally insert more details (if needed).

**2.2.2 Model set-up**

L166: Re-word from "...not influenced..." to "...not directly influenced..."

As a suggestion, you may also summarize this paragraph (not as much as the previous two) and insert some more descriptive information in Annex.

**Table 1:** Consider moving this table to Annex.

Figure 3: Consider moving this figure to Annex.

Table 2: Consider moving this table to Annex.

**Reservoirs set-up**

**Table 3:** Consider moving this table to Annex. Additionally, consider inserting the minimum and maximum volume, and the minimum outflow values inside Figure 4. Hence, this information would be kept in the main text even though the insertion of the table in annex.

**Figure 4:** Change the title to "Level *versus* stored volume curves for (a) Portodemouros, (b) Bandariz, and (c) Touro reservoirs." Include the numbering (a), (b) and (c) outside the figure at the right upper part, as it was in Figure 2.

L211: Delete the sentence "and by default".

**2.3 Neural network model for reservoir outflow estimation**

L217-218: Delete "...(convolutional neural network, long short-term memory, multi-layer perceptron, extreme learning machine, etc.)"

L234-236: Consider the use of a connector for the beginning of the paragraph (e.g., additionally) and rephrase the sentence to a non-passive voice.

L234-240: Consider rephrasing this paragraph for clarification.

**2.3.1 Input data**

L242: Delete: "selected from a set that included".

**2.3.2 Structure**

L255: Delete "based".

L256-257: Delete: "As referred before, a CLSTM model is based on convolutional and long short-term memory layers."

**Figure 5:** Change the title to "CLSTM structure used in this work". Please consider showing also the variables in this scheme, or the possibility of merging this figure with Figure 6.

L262: Insert "the" before "convolutional layers", i.e., "For the convolutional layers..."

L265: Please consider rephrasing to: "...The loss was estimated using the mean absolute error (MAE). Finally, the number of epochs and the batch size were respectively 300 and 20, found after a trial and error procedure."

**2.3.3 Model optimization**

L270-272: Please consider to do not be so descriptive about the packages and delete these sentences: "The training dataset was handled and prepared with Pandas (McKinney, 2010) and Scikit-learn Pedregosa et al., 2011) packages, with the data being delayed with the first and scaled with the latter."

L272: Rephrase to: "The data was scaled using Eq. 1 to each variable independently."

L275-277: Change the variables to the italic format in the text.

**2.4 Coupling MOHID-Land and CLSTM models**

L294-296: Rewrite the sentence using a non-passive voice.

L300: "...used to stabilize the hydrological model," was already mentioned before.

L303-304: Re-word from "This model" to "The calibrated model". Add the plural in "level" and "inflow".

L307: Delete: "This scheme was coded in the Python language.".

Figure 6: As mentioned before, please consider the possibility of merging this figure with Figure 5.

**2.5 Model's evaluation**

The text in this section must be re-organized for clarity. The content is all there, but the sentences seem out of order, confusing and repetitive. Please consider the following structure: (a) first you give the initial idea (as done), (b) then you present and discuss the validation period; (c) then you present and discuss the test period; (d) finally you explain the metrics used. The metrics were used for both periods; therefore, it would sound better if they were mentioned after or before the two periods, and not between them.

L314: Re-word from "analysis" to "inspection"; and "namely" to ",i.e.,".

L316: Re-phrase to: "...which were computed using Eqs 2-5 respectively."

L323: Insert a new paragraph at "The test dataset..."

L330: Consider the new work from Moriasi as an updated reference: Moriasi DN, Gitau MW, Pai N, Daggupati P (2015) Hydrologic and water quality models: Performance measures and evaluation criteria. Trans ASABE (am Soc Agric Biol Eng) 58(6):1763–1785. https://doi.org/10.13031/trans.58.10715

L330: Correct from "R2" to "R2".

**3 Results**

**3.1 MOHID-Land model**

I would suggest the authors to consider mentioning Figure 7 on section 3.3 instead that at this section.

**Table 4:** Please consider the insertion of a new column referring to the location of the station, i.e., upstream of downstream the reservoirs.

L341: After ending the discussion about the Sar, Ulla, Arnego-Ulla and Deza hydrometric stations, please consider opening a small description of the results from Table 4 for Ulla-Touro and Ulla-Theo. Please ensure to mention in the text the potential causes of the low-performance.

**Figure 7:** Besides moving this figure to section 3.3, please consider adjusting the y-axis range of the two subplots to the same scale. I suggest both to "0-600". It is always better for a visual comparison from the readers. Move the labels "a" and "b" to the upper left outside of the subplots area.

**3.2 CLSTM model**

**Table 5:** Add a row with the chosen set of statistical variables. You have the average, minimum, maximum, and standard deviation, but an additional row with the chosen set would be useful.

L355: Delete "always".

L359: Consider to add for clarification: "...outflow using observed levels and inflows as forcing."

L364: You observed May and June 2016 in Figure 8. Can this period be highlighted in the figure?

**Figure 8:** Please consider rephrasing the title to "Comparison between modelled and observed Portodemouros outflow considering the CLSTM model."

**3.3 Coupled system**

Figures 8 and 9: Please consider to merge those two figures in a single one.

**Figure 9:** Change the title to "Comparison between the modelled and observed (a) inflow and (b) outflow in Portodemouros reservoir using the coupled model."

Figure 10: Please consider moving this figure to the discussion section.

L395: Delete: "It could be expected that this issue would affect streamflow estimation downstream the reservoir since the outflow estimated by CLSTM model considered the level values estimated by MOHID-Land." Please consider deleting this phrase and keeping just the last sentence.

**3.4 Impact of reservoirs 'operation on streamflow**

L403-404: Rephrase to "with reservoirs (Res.)" and "without reservoirs (No res.)". The abbreviation should be placed after the words.

**4 Discussion**

**L451-454:** The sentence: "With the choice of the forcing variables being pointed out by several authors as crucial for a successful model (ASCE, 1996; Maier et al., 2010; Dolling and Varas, 2002; Wu et al., 2014; Juan et al., 2017), the consideration of other forcing variables should be evaluated." Seems disconnect from the previous sentence. Please consider rephrasing it.

L487: Did you mean by "environmentally friendly streamflow" the "environmental/ecological flow"? If yes, please rephrase it.

**5** Conclusion**

L529: As you did not do the test I would consider changing this word from "...in part..." to "...probably...".

L529: Consider the inclusion of: "... on the performance of the coupled system in the computation of daily streamflow...".

**References:**

Oliveira, A. R., Ramos, T. B., Simionesei, L., Pinto, L., and Neves, R.: Sensitivity Analysis of the MOHID-Land Hydrological Model: A Case Study of the Ulla River Basin. Water, 12(11), 3258, https://doi.org/10.3390/w12113258, 2020.

Moriasi DN, Gitau MW, Pai N, Daggupati P (2015) Hydrologic and water quality models: Performance measures and evaluation criteria. Trans ASABE (am Soc Agric Biol Eng) 58(6):1763–1785. https://doi.org/10.13031/trans.58.10715

---

## Author Comment (AC2)

Dear editor and reviewers,

Thank you for your constructive comments and suggestions about our manuscript. We revised the manuscript taking into account your suggestions and comments. Please find attached a point-by-point response, with our answers in blue. We hope that the revised version of the manuscript properly addresses your concerns.

Sincerely,

Ana Oliveira on behalf of all authors

**Author's responses to reviewer #1**

**Referee #1 - Thiago Victor Medeiros do Nascimento**
**General comments**

The manuscript presents an interesting and relevant work addressing scientific questions within the scope of HESS. The authors present a successful newly implemented coupled system with the main objective of estimating the outflow from a reservoir in Spain. The methodology developed and applied merges the physical understanding of a physical-based model and the capabilities of artificial neural networks to learn from non-linear processes. Therefore, applying a coupled system highlights the novelty of the work, which is not simply another application of a hydrological model or an artificial neural network to estimate streamflow in a given catchment. In this sense, the authors gave proper credit to related work and clearly indicated their original contribution. The manuscript is well-structured, clear and concise. It is well-written and easy to understand. The authors present a good introduction of state of the art regarding the subject and a well-written description of the study area and the materials ad methods applied. The studies used for introducing the subject and the discussion are relevant and recent. The results presentation and discussion gave a clear and sufficient overview of the results and their respective limitations. The conclusions present an interesting closing of the presented workflow concisely and straightforwardly, but not forgetting to present the future perspectives from this work. Finally, the authors present the workflow code in a public and open repository, contributing significantly to open science and making it possible the work reproducibility. Therefore, the paper deserves to be published in HESS after some corrections and adaptations regarding the quality of the writing and figures.

We appreciate the positive comments of referee #1 and the interest about the work presented in the manuscript. We would like to thank referee #1 for its time to analyze the manuscript.

**Specific comments**

In this section, I present some specific comments to be answered and implemented in the manuscript. No changes in the methodology are proposed, only suggestions for the writing, discussion and overall figures and tables quality.

1. The authors provided a complete public repository with the workflow. I recommend not going into too much detail within the text regarding the specific Python libraries and functions applied in this methodology.

The methodology section was written to allow the reproducibility of the work and to give the credits to the authors that previously developed the tools involved in this study. In that sense, we decided to detail the neural network development and cite the authors of the packages used here. Nevertheless, we agree that in some cases the description doesn't need to be so detailed. Thus, the manuscript was modified as follows:

*Old version:*

"The scaling function was the "MinMaxScaler", which applies equation 1 to each variable in the dataset independently:

$$x_{scaled} = \frac{(x - x_{min})}{x_{max} - x_{min}}(M - m) + m, \tag{1}$$

where $x_{scaled}$ is the scaled value, x is the original value, $x_{max}$ and $x_{min}$ are the maximum and minimum values of the variable being scaled, and M and m are the maximum and minimum values of the desired range of the scaled data. Considering that the maximum values of the variables cannot be represented in the dataset, the desired range was defined from 0 to 0.9."

*New version:*

"The scaling function transformed the features of the given data into a value within a desired range, which was defined from 0 to 0.9 considering that the maximum values of the variables cannot be represented in the dataset."

2. Likewise, regarding the MOHID-Land model, since the model was already implemented in the referenced work of Oliveira et al. (2020) where it was fully described, the information regarding the implementation not already presented in this work should be given in the present details in Annex instead of in the main text. The authors can consider only a summary of the information instead of being in such details in the present work.

The description of MOHID-Land model is already summarized in this manuscript, which can be confirmed when compared with the description presented in Oliveira et al. (2020). The reservoirs module is described for the first time in this manuscript, which justifies being placed in the main text. The model set-up was summarized, and the details were moved to Appendix A. Thus, in the new version the model set-up section will be as follows:

"The MOHID-Land model was already implemented, calibrated, and validated in the study area as detailed in Oliveira et al. (2020). This study was carried out from 01/01/2008 to 31/12/2017. Only the natural regime flow in the watershed was considered, with model calibration and validation using data from hydrometric stations not influenced by reservoirs' operations. A detailed description of the calibrated parameters resulting from the work done by Oliveira et al. (2020) is presented in Appendix A."

3. The authors conclude that the poor representation of reservoir levels reflects the non-inclusion of evaporative losses in the model (L529). I agree, but do the authors have concrete results corroborating this? Did you perform tests including it in your model? Would that be a possibility for further work? For this work, it is optional to make this inclusion. However, more evidence could improve your conclusions.

The current version of MOHID-Land model does not allow the calculation of evaporation losses in reservoirs. It would be necessary to program the respective subroutines/functions to include those losses in the simulation. Thus, no tests were performed related to that. Nevertheless, we expect to include the simulation of reservoir's evaporation losses in MOHID-Land code soon.

4. I would like to have a deeper discussion about why, even with the limitation in reproducing the reservoir levels, the model still performed well in reproducing the streamflow at the outlet. Is this due somehow to the concentration time of the watershed? A slight increase in the discussion will enrich your paper.

The following was added in L473:

"This is intimately related with the discussion presented by Kirchner (2006) about obtaining the right results for the right reasons, and where the author explores the limitations of the operational practice of hydrology. In that sense, the coupled system presented here, namely the CSLTM model, seems to be obtaining the right answer but for the wrong reasons. With the behavior of CLSTM model being classified as a "black box", without any physical constraints implemented, its results can be good enough while the model is exposed to conditions similar to those used for its optimization. However, when the forcing conditions go far beyond those used in the optimization, the results of these type of models become unreliable because of their lack of physical realism."

5. L361: What would cause this behavior by the model in your option? Could it be further discussed in the text?

The behavior commented in L361 (Results section) is a direct result of the CLSTM, which the viability and limitations are then discussed in the Discussion section L451-461.

6. L452: In this part of the discussion, the authors claim several limitations to using the CLSTM model. The text follows, describing several concerns, but are those the limitations intended by the authors to be mentioned here? I recommend rewriting this part text for clarification.

Following the previous comment, in this part of the text authors intend to discuss what can be the limitations of the CLSTM model developed in the present study. This discussion is made because the results are not perfect as demonstrated in L361.

7. L459-461: The authors claim that the optimization of the CLSTM could improve the results. Why was this not tested for this work? I would appreciate a further discussion of this matter in the manuscript text.

We are proposing a methodology to improve streamflow calculation in watersheds where the natural regime flow is altered by the operation of reservoirs. We proposed the methodology, tested it, and presented results that are totally acceptable demonstrating the adequacy of the methodology. In that sense, we consider that we fulfilled the objective of the study, and the results are in line with other studies, as presented in the discussion section.

**Technical corrections**

In this section, I present some technical corrections and suggestions to be implemented in the text. This section is divided into an initial general part and later divided by each text section (e.g., Abstract, Introduction).

**General:**

Please refer to the guidelines template available at (https://www.hydrology-and-earth-system-sciences.net/submission.html#templates) when reviewing your manuscript.

**Symbols and variables:**

Consider a full check in all the symbols, such as "m", "Km" etc., to ensure they are correctly placed near their respective words. Consider using a standard number of decimals in nearby sentences, e.g., "11.6 and 2.5m", and not "11.6 and 2.55m".

*It was revised and changed in the new version.*

**Equations and variables:**

For all equations, please make sure to cite them in the text before presenting them using the HESS guidelines for equations, i.e., using the label "Eq. (X)" or "Equation (X)" depending on the situation. It is a better practice to refer to your equation as:"…using Eq. (X):" instead of solely mention "…using the equation below:". Additionally, Make sure to have all variables in italics when mentioned in the full text, e.g., $x_{scaled}$ and not xscaled.

*It was revised and changed in the new version.*

**Figures and tables:**

Please consider improving the figures and tables' quality regarding font size, style and colors. The figures should have a similar style, and the authors can think of improving their quality following some suggestions in the further sections. Please ensure that all figures and tables located in the main text are mentioned in the text. Finally, please move some tables and figures to the annexes and reduce the final number of figures in the manuscript's main text by merging them when possible.

**Abstract**

L10: Re-word from "estimate" to "estimation".

*L10: authors opted for maintain the word estimate. It seems more appropriate.*

L12: Please present the main objective of the manuscript more clearly. Beginning the presentation of your objective in the abstract with: "In this study, and already implemented…" does not capture the main objective of this study which is the development and application of a coupled system. Please start your objective presentation by making sure to present this information. Make sure to have your abstract following this pathway: Contextualization/Relevance – Objective – Methodology – Results – Conclusions.

L13: Please consider deleting the part: "…, calibrated, and validated solution of…"

*L12 and L13: The text was revised as follows:*

*Old version:*

"In this study, an already implemented, calibrated, and validated solution of MOHID-Land model for natural regime flow in Ulla River basin was considered as baseline."

*New version:*

"In this study, an already implemented solution of MOHID-Land model for natural regime flow in Ulla River basin was considered as baseline."

L18: Re-word from "This coupled system was daily evaluated in two hydrometric stations…" to "This coupled system was evaluated on a daily basis using two hydrometric stations…".

L18: Done.

L22: Consider changing "modified rivers" to "modified catchments".

L22: Done.

**1 Introduction**

L24-28: Consider to summarize this list of activities and situations.

L24-28: we have considered the reviewer's comment but decided to maintain the text as it is.

L31: Delete "mechanisms". It was already mentioned and it is redundant. Forcing is enough.

L31: The new version was modified as follows:

*Old version:*

"These non-linear forcing mechanisms include meteorological conditions, land use, infiltration, morphological features of the river, and catchment characteristics (Mohammadi et al., 2021)."

*New version:*

"These non-linear forcings include meteorological conditions, land use, infiltration, morphological features of the river, and catchment characteristics (Mohammadi et al., 2021)."

L37: Re-word from "makes it impeditive" to "poses a challenge".

L37: The new version was modified as follows:

*Old version:*

"If hydrological models are prepared to study and comprehend the behavior of natural systems, the lack of information about reservoirs' operations such as operating rules and flood contingency plans makes it impeditive for a correct representation of those infrastructures."

*New version:*

"If hydrological models are prepared to study and comprehend the behavior of natural systems, the lack of information about reservoirs' operations such as operating rules and flood contingency plans poses a challenge for a correct representation of those infrastructures."

L42: Delete the extra space after "faced".

L42: Done.

L54-56: Consider using numbering to list those two tests, i.e., "(a) first the weir was simulated as static (with closed sluice) to identify the sluice 55 operating rules by comparing results with the known outflow time series; (b) second the weir model was run as non-static to implement the sluice operating rules deducted from the first approach."

L54-56: The new version was modified as follows:

*Old version:*

"The authors considered a weir model and two tests were performed: first the weir was simulated as static (with closed sluice) to identify the sluice operating rules by comparing results with the known outflow timeseries; second the weir model was run as non-static to implement the sluice operating rules deducted from the first approach."

*New version:*

"The authors considered a weir model and two tests were performed: (i) the weir was simulated as static (with closed sluice) to identify the sluice operating rules by comparing results with the known outflow timeseries; (ii) the weir model was run as non-static to implement the sluice operating rules deducted from the first approach."

L56: Consider start a new paragraph after the previous mentioned statements.

L56: we have considered the reviewer's comment but decided to maintain the text as it is.

L76: I believe that if you keep using "the physical-based…" you should refer the official MOHID-Land reference, and not the previous work. However, if you re-word to "a physical based distributed Mohid-Land…" you may keep the reference to your previous work.

L76: MOHID-Land does not have an official reference. Nevertheless, we changed the sentence as follows:

*Old version:*

"In the present study, the physical-based, distributed MOHID-Land model (Oliveira et al., 2020) was coupled with a Convolutional Long Short-Term Memory (CLSTM) model to estimate the daily outflow in Portodemouros reservoir, Galicia, Spain."

*New version:*

"In the present study, the physical-based, distributed MOHID-Land model (Trancoso et al. 2009, Canuto et al., 2019, Oliveira et al., 2020) was coupled with a Convolutional Long Short-Term Memory (CLSTM) model to estimate the daily outflow in Portodemouros reservoir, Galicia, Spain."

L80: Consider moving "However, the CLSTM model was first trained and tested using historical data." to L78, after you introduced the CLSTM model.

L78-80: This part of the text intends, firstly, to present the composition of the coupled system and briefly explain how the data exchange was implemented. Then, a focus on the CLSTM model is made to explain that it was trained and tested with observed data instead of the data becoming from MOHID-Land model.

We don't see the necessity of changing the order of the explanation since it starts from the general information and then goes to a small detail of one of the parts of the general description.

**2 Materials and methods**

**2.1 Description of the study area**

L90: Consider re-word from: "-0.75m" to "-1m". Is it not necessary to be so precise at this description.

L90: revised based on the reviewer's comments.

L92: Insert a reference for the Köppeb-Geiger classification.

L92: Reference was added here and in the list of references.

L102: Clarify what you mean by "set of reservoirs". The other two reservoirs? Or the whole set?

L102: The sentence was modified as follows:

*Old version:*

"Due to its significative storing capacity, Portodemouros reservoir can be used for flood control, however, the set of reservoirs is mainly responsible for energy production."

*New version:*

"Due to its significative storing capacity, Portodemouros reservoir can be used for flood control, however, the set of three reservoirs is mainly responsible for energy production."

L104: Consider to use a synonym for "however" since it was already mentioned nearby.

L104: The sentence was modified as follows:

*Old version:*

"However, Portodemouros works in a different way, presenting significative differences between the inflow and outflow patterns (Figure 2a and d)."

*New version:*

"Nevertheless, Portodemouros works in a different way, presenting significative differences between the inflow and outflow patterns (Figure 2a and d)."

**Figure 1:** Consider inserting the "Atlantic ocean" and "Mediterranean sea" labels on the figure. The palette choice for the countries is not colorblind friendly (consider alternative colors for green and red together). It is also a little difficult to identify the reservoirs in the figure. Consider altering the font sizes with their respective names, or moving these labels. Re-word "water bodies/lines" to "water courses". Re-word "Basin delineation" to "Ulla river watershed" for clarity. Check the legend color bar labels. Are those correct? Finally, it is missing the latitude and longitude or X and Y coordinates of the study area. They should be inserted in the map.

Figure 1: the figure was revised based on the reviewer's comments.

[Figure]

**Figure 2:** You have four subplots here. I would consider two changes: (1) subplots a-c should have the same Y-scale, e.g., 0-450 $m^3s^{-1}$. Subplot d can keep its scale since it refers to a different time-period.

Figure 2: Done.

[Figure]

**2.2 MOHID-Land description & 2.2.1 Reservoirs module**

As this work uses the already implemented model from Oliveira et al. (2020), I would consider a summarization of these sections, and if needed the addition of this full description in Annex. This description is interesting to understand the MOHID-Land implementation; however, it is too long for this section and for the scope of the present work. In Annex, you can additionally insert more details (if needed).

**2.2.2 Model set-up**

L166: Re-word from "…not influenced…" to "…not directly influenced…"

L166: we have considered the reviewer's comment but decided to maintain the text as it is, since the stations are not influenced by the reservoirs. The referred hydrometric stations are placed upstream the reservoirs, far away from the possible influence of the flooded area of Portodemouros reservoir. Thus, those stations are not directly or indirectly influenced by reservoirs' operation.

As a suggestion, you may also summarize this paragraph (not as much as the previous two) and insert some more descriptive information in Annex.

**Table 1:** Consider moving this table to Annex.

**Figure 3:** Consider moving this figure to Annex.

**Table 2:** Consider moving this table to Annex.

Most of the information presented in this sub-section was moved to Annex as suggested by the referee. Table 1, Figure 3 and Table 2 were also moved to Annex.

**Reservoirs set-up**

**Table 3:** Consider moving this table to Annex. Additionally, consider inserting the minimum and maximum volume, and the minimum outflow values inside Figure 4. Hence, this information would be kept in the main text even though the insertion of the table in annex.

Table 3: We appreciate the suggestion but since this information is related with the additional implementation made for this study, we think that for a better understanding the table should remain in the main text.

**Figure 4:** Change the title to "Level versus stored volume curves for (a) Portodemouros, (b) Bandariz, and (c) Touro reservoirs." Include the numbering (a), (b) and (c) outside the figure at the right upper part, as it was in Figure 2.

Figure 4: Done.

L211: Delete the sentence "and by default".

L211: Done.

**2.3 Neural network model for reservoir outflow estimation**

L217-218: Delete "...(convolutional neural network, long short-term memory, multi-layer perceptron, extreme learning machine, etc.)"

L217-218: Done.

L234-236: Consider the use of a connector for the beginning of the paragraph (e.g., additionally) and rephrase the sentence to a non-passive voice.

L234-240: Consider rephrasing this paragraph for clarification.

L234-236 and L234-240: we have considered the reviewer's comment but decided to maintain the text as it is.

**2.3.1 Input data**

L242: Delete: "selected from a set that included".

L242: We believe that if we delete this part of the sentence, it will not make sense, since we intend to present the list of variables that we tested as forcing variables. From that set, only the inflow and level effectively remained as forcing variables.

**2.3.2 Structure**

L255: Delete "based".

L255: The sentence was modified as follows:

*Old version:*

"In this study, the model structure was developed based on Python language and using Keras package (Chollet et al., 2015), on top of TensorFlow (Abadi et al., 2016)."

*New version:*

"In this study, the model structure was developed using Python language and Keras package (Chollet et al., 2015), on top of TensorFlow (Abadi et al., 2016)."

L256-257: Delete: "As referred before, a CLSTM model is based on convolutional and long short-term memory layers."

L256-257: Done.

**Figure 5:** Change the title to "CLSTM structure used in this work". Please consider showing also the variables in this scheme, or the possibility of merging this figure with Figure 6.

Figure 5: Done.

[Figure]

Figure 4 CLSTM structure used in this study.

L262: Insert "the" before "convolutional layers", i.e., "For the convolutional layers…"

L262: Done.

L265: Please consider rephrasing to: "…The loss was estimated using the mean absolute error (MAE). Finally, the number of epochs and the batch size were respectively 300 and 20, found after a trial and error procedure."

L265: Done.

**2.3.3 Model optimization**

L270-272: Please consider to do not be so descriptive about the packages and delete these sentences: "The training dataset was handled and prepared with Pandas (McKinney, 2010) and Scikit-learn Pedregosa et al., 2011) packages, with the data being delayed with the first and scaled with the latter."

L272: Rephrase to: "The data was scaled using Eq. 1 to each variable independently."

L275-277: Change the variables to the italic format in the text.

L270-277: As referred to in the Specific comments, we prefer to maintain the reference to the used packages and the citation of the authors of those packages. It will allow an easier reproduction of this work for the interested readers. However, to avoid the detailed description referred by the referee, we

deleted Eq. 1 and the corresponding explanation, since it can be easily found online in the webpage of the respective package.

**2.4 Coupling MOHID-Land and CLSTM models**

L294-296: Rewrite the sentence using a non-passive voice.

L294-296: we have considered the reviewer's comment but decided to maintain the text as it is.

L300: "…used to stabilize the hydrological model," was already mentioned before.

L300: the sentence was rewritten as follows:

*Old version:*

"In the warm-up simulation, used to stabilize the hydrological model, the  reservoirs' module was deactivated."

*New version:*

"In the warm-up simulation the  reservoirs' module was deactivated."

L303-304: Re-word from "This model" to "The calibrated model". Add the plural in "level" and "inflow".

L303-304: We can't re-word as referee suggests since the CLSTM was not calibrated but optimized. Thus, the sentence was modified as follows:

*Old version:*

"This model was loaded with the weights already optimized and received the information about the level and the inflow of Portodemouros reservoir estimated by MOHID-Land for the three days before the simulated day."

*New version:*

"The optimized model was loaded and received the information about the levels and the inflows of Portodemouros reservoir estimated by MOHID-Land for the three days before the simulated day."

L307: Delete: "This scheme was coded in the Python language.".

L307: Done.

**Figure 6:** As mentioned before, please consider the possibility of merging this figure with Figure 5.

Figure 6: we appreciated the reviewer's comment but we think that separated figures are more easier to understand for the reader.

**2.5 Model's evaluation**

The text in this section must be re-organized for clarity. The content is all there, but the sentences seem out of order, confusing and repetitive. Please consider the following structure: (a) first you give the initial idea (as done), (b) then you present and discuss the validation period; (c) then you present and discuss the test period; (d) finally you explain the metrics used. The metrics were used for both periods; therefore, it would sound better if they were mentioned after or before the two periods, and not between them.

The section was modified according to referee suggestions and as follows:

*Old version:*

"The CLSTM model used to predict the outflow from Portodemouros reservoir was evaluated considering a subset of the original dataset from Augas de Galicia, which was not previously exposed to the trained model. That subset is known as test dataset and contained pairs of forcing (inflow and level) and target (outflow) variables. Thus, the outflow was estimated based on the forcing variables and was then compared to the corresponding measured outflow. This comparison was based on a visual analysis, and the estimation of four different statistical indicators, namely, the coefficient of determination ($R^2$), the percentage bias (PBIAS), the ratio of the root mean square error to the standard deviation of observation (RSR), and the Nash-Sutcliffe modeling efficiency (NSE), which were computed as follows:

$$R^2 = \left[ \frac{\sum_{i=1}^{p}(X_i^{obs}-X_{mean}^{obs})(X_i^{sim}-X_{mean}^{sim})}{\sqrt{\sum_{i=1}^{p}(X_i^{obs}-X_{mean}^{obs})^2}\sqrt{\sum_{i=1}^{p}(X_i^{sim}-X_{mean}^{sim})^2}} \right]^2, \tag{2}$$

$$PBIAS = \frac{\sum_{i=1}^{p}(X_i^{obs}-X_i^{sim})}{\sum_{i=1}^{p}X_i^{obs}} \times 100, \tag{3}$$

$$RSR = \frac{RMSE}{STDEV_{obs}} = \frac{\sqrt{\sum_{i=1}^{p}(X_i^{obs}-X_i^{sim})^2}}{\sqrt{\sum_{i=1}^{p}(X_i^{obs}-X_{mean}^{obs})^2}}, \tag{4}$$

$$NSE = 1 - \frac{\sum_{i=1}^{p}(X_i^{obs}-X_i^{sim})^2}{\sum_{i=1}^{p}(X_i^{obs}-X_{mean}^{obs})^2}, \tag{5}$$

where $x_i^{obs}$ and $x_i^{sim}$ are the outflow values observed and estimated by the model on day i, respectively, $X_{mean}^{obs}$ and $X_{mean}^{sim}$ are the average outflow considering the observed and the modelled values in the analyzed period, and p is the total number of days/values in this period. The test dataset corresponded to 10% of the size of the original dataset and covered the period between 19/09/2015 and 16/07/2018, totalizing 1023 daily values.

In this study the evaluation of streamflow values focused the hydrometric stations placed downstream the set of reservoirs and intended to verify the behavior of the coupled modelling system (MOHID-Land+CLSTM). This evaluation was performed by comparing the streamflow values estimated by the coupled modelling system with those measured in Ulla-Touro and Ulla-Teo hydrometric stations. The validation of the coupled system was made from 01/01/2009 to 31/12/2017 and was based on a visual analysis and the four statistical indicators presented before, namely, the $R^2$, PBIAS, NSE, and RSR. According to Moriasi et al., 2007, the NSE and the R2 values should be higher than 0.5 and the PBIAS should be ±25% for the model performance to be considered satisfactory, while RSR values closer to 0 mean a more accurate model."

*New version:*

"The CLSTM model used to predict the outflow from Portodemouros reservoir was evaluated considering a subset of the original dataset from Augas de Galicia, which was not previously exposed to the trained model. That subset is known as test dataset and contained pairs of forcing (inflow and level) and target (outflow) variables. Thus, the outflow was estimated based on the forcing variables and was

then compared to the corresponding measured outflow. The test dataset corresponded to 10% of the size of the original dataset and covered the period between 19/09/2015 and 16/07/2018, totalizing 1023 daily values.

In turn, the evaluation of streamflow values focused the hydrometric stations placed downstream the set of reservoirs and intended to verify the behavior of the coupled modelling system (MOHID-Land+CLSTM). This evaluation was performed by comparing the streamflow values estimated by the coupled modelling system with those measured in Ulla-Touro and Ulla-Teo hydrometric stations. The validation of the coupled system was made from 01/01/2009 to 31/12/2017.

In both cases, the comparison between modelled and observed values was based on a visual inspection, and the estimation of four different statistical indicators, namely, the coefficient of determination ($R^2$), the percentage bias (PBIAS), the ratio of the root mean square error to the standard deviation of observation (RSR), and the Nash-Sutcliffe modeling efficiency (NSE), which were computed using Eqs 1-4, respectively.

$$R^2 = \left[ \frac{\sum_{i=1}^{p}(X_i^{obs}-X_{mean}^{obs})(X_i^{sim}-X_{mean}^{sim})}{\sqrt{\sum_{i=1}^{p}(X_i^{obs}-X_{mean}^{obs})^2}\sqrt{\sum_{i=1}^{p}(X_i^{sim}-X_{mean}^{sim})^2}} \right]^2, \tag{1}$$

$$PBIAS = \frac{\sum_{i=1}^{p}(X_i^{obs}-X_i^{sim})}{\sum_{i=1}^{p} X_i^{obs}} \times 100, \tag{2}$$

$$RSR = \frac{RMSE}{STDEV_{obs}} = \frac{\sqrt{\sum_{i=1}^{p}(X_i^{obs}-X_i^{sim})^2}}{\sqrt{\sum_{i=1}^{p}(X_i^{obs}-X_{mean}^{obs})^2}}, \tag{3}$$

$$NSE = 1 - \frac{\sum_{i=1}^{p}(X_i^{obs}-X_i^{sim})^2}{\sum_{i=1}^{p}(X_i^{obs}-X_{mean}^{obs})^2}, \tag{4}$$

where $x_i^{obs}$ and $x_i^{sim}$ are the outflow values observed and estimated by the model on day $i$, respectively, $X_{mean}^{obs}$ and $X_{mean}^{sim}$ are the average outflow considering the observed and the modelled values in the analyzed period, and $p$ is the total number of days/values in this period.

According to Moriasi et al., 2015, the NSE must be higher than 0.50 for the model to be classified as satisfactory, higher than 0.70 to be good and higher than 0.8 for a very good performance. The $R^2$ values should be higher than 0.60 for a satisfactory performance, higher than 0.75 for good behavior and higher than 0.85 to be classified as very good. Finally, PBIAS of ±5% is a characteristic of a very good model, while a model with a PBIAS of ±10% is classified as good. To be classified as satisfactory, model's PBIAS should be ±15%."

L314: Re-word from "analysis" to "inspection"; and "namely" to ",i.e.,".

L314: "Analysis" was replaced by "inspection". However, we appreciate the suggestion, but we prefer to maintain the "namely" instead of "i. e.", since the latter means that is and it seems do not correctly fit in the sentence.

L316: Re-phrase to: "…which were computed using Eqs 2-5 respectively."

L316: Done.

L323: Insert a new paragraph at "The test dataset…"

L323: Done.

L330: Consider the new work from Moriasi as an updated reference: Moriasi DN, Gitau MW, Pai N, Daggupati P (2015) Hydrologic and water quality models: Performance measures and evaluation criteria. Trans ASABE (am Soc Agric Biol Eng) 58(6):1763–1785. https://doi.org/10.13031/trans.58.10715

L330: Correct from "R2" to "$R^2$".

L330: Done.

**3 Results**

**3.1 MOHID-Land model**

I would suggest the authors to consider mentioning Figure 7 on section 3.3 instead that at this section.

**Table 4:** Please consider the insertion of a new column referring to the location of the station, i.e., upstream of downstream the reservoirs.

Table 4: Modified as follows:

| Station | $R^2$ (-) | | NSE (-) | | RSR (-) | | PBIAS (%) | | Position relative to reservoirs |
|---|---|---|---|---|---|---|---|---|---|
| | Cal. | Val. | Cal. | Val. | Cal. | Val. | Cal. | Val. | |
| Sar | 0.75 | 0.83 | 0.72 | 0.81 | 0.53 | 0.44 | 0.18 | 16.09 | |
| Ulla | 0.56 | 0.76 | 0.55 | 0.72 | 0.67 | 0.53 | -11.24 | -18.54 | Upstream |
| Arnego-Ulla | 0.70 | 0.78 | 0.69 | 0.76 | 0.55 | 0.49 | -12.29 | -16.82 | |
| Deza | 0.74 | 0.85 | 0.72 | 0.84 | 0.53 | 0.40 | -8.96 | -4.35 | |
| Ulla-Touro | 0.46 | 0.52 | -0.09 | 0.24 | 1.04 | 0.87 | -19.06 | -19.12 | Downstream |
| Ulla-Teo | 0.77 | 0.79 | 0.71 | 0.73 | 0.54 | 0.52 | -16.68 | -14.36 | |

**Table 2 Statistical indicators resulting from the comparison of the natural regime flow estimated by MOHID-Land with the observed streamflow values in 6 hydrometric stations (Cal. – calibration, Val. – validation, adapted from: Oliveira et al., 2020).**

L341: After ending the discussion about the Sar, Ulla, Arnego-Ulla and Deza hydrometric stations, please consider opening a small description of the results from Table 4 for Ulla-Touro and Ulla-Theo. Please ensure to mention in the text the potential causes of the low-performance.

L341: We appreciate the referee's comments but, with this section being dedicated to present the results and the improvement of the streamflow values in the two stations being the main goal of this work, the authors think that no more discussion is needed here about Ulla-Touro and Ulla-Teo hydrometric stations.

Figure 7: Besides moving this figure to section 3.3, please consider adjusting the y-axis range of the two subplots to the same scale. I suggest both to "0-600". It is always better for a visual comparison from the readers. Move the labels "a" and "b" to the upper left outside of the subplots area.

Figure 7: the authors placed this figure in this section because the results of the MOHID-Land model running for natural regime flow are presented here. Thus, for the authors, it makes sense to refer and present this figure here to show how that version of the hydrological model (natural regime flow) fits the observed values. Nonetheless, the Figure was modified according to the comments of referee #2:

[Figure]

**Figure 7 Comparison of modelled and observed average monthly streamflow in hydrometric stations (a) Ulla-Touro and (b) Ulla-Teo with and without considering the existence of reservoirs. Focus on the daily values for the period between September 2013 and September 2014 in (c) Ulla-Touro and (d) Ulla-Teo hydrometric stations.**

**3.2 CLSTM model**

**Table 5:** Add a row with the chosen set of statistical variables. You have the average, minimum, maximum, and standard deviation, but an additional row with the chosen set would be useful.

Table 5: Table modified as follows:

|  | $R^2$ (-) | NSE (-) | RSR (-) | PBIAS (%) |
|---|---|---|---|---|
| **Average** | 0.90 | 0.89 | 0.33 | -1.71 |
| **Minimum** | 0.89 | 0.86 | 0.31 | -15.74 |
| **Maximum** | 0.91 | 0.90 | 0.37 | 14.07 |
| **Standard deviation** | 0.00 | 0.01 | 0.01 | 6.26 |
| **Elected model** | 0.91 | 0.90 | 0.31 | 2.61 |

**Table 3 Average, minimum, maximum and standard deviation values of the four statistical parameters estimated for the set of 100 models ran and the elected model.**

L355: Delete "always".

L355: Done.

L359: Consider to add for clarification: "…outflow using observed levels and inflows as forcing."

L359: Done.

L364: You observed May and June 2016 in Figure 8. Can this period be highlighted in the figure?

L364: Done.

**Figure 8:** Please consider rephrasing the title to "Comparison between modelled and observed Portodemouros outflow considering the CLSTM model."

Figure 8: Done.

**3.3 Coupled system**

**Figures 8 and 9:** Please consider to merge those two figures in a single one.

Figure 8 and 9: we appreciated the reviewer's comment, but we think that separated figures are more easier to understand for the reader.

**Figure 9:** Change the title to "Comparison between the modelled and observed (a) inflow and (b) outflow in Portodemouros reservoir using the coupled model."

Figure 9: Done.

**Figure 10:** Please consider moving this figure to the discussion section.

Figure 10: The modelled level is a result of the coupled system. Thus, in our opinion it should be presented here to be discussed in the Discussion section.

L395: Delete: "It could be expected that this issue would affect streamflow estimation downstream the reservoir since the outflow estimated by CLSTM model considered the level values estimated by MOHID-Land." Please consider deleting this phrase and keeping just the last sentence.

L395: we appreciated the reviewer's comment but we do not understand why we should delete the sentence.

**3.4 Impact of reservoirs 'operation on streamflow**

L403-404: Rephrase to "with reservoirs (Res.)" and "without reservoirs (No res.)". The abbreviation should be placed after the words.

L403-404: Done.

**4 Discussion**

L451-454: The sentence: "With the choice of the forcing variables being pointed out by several authors as crucial for a successful model (ASCE, 1996; Maier et al., 2010; Dolling and Varas, 2002; Wu et al., 2014; Juan et al., 2017), the consideration of other forcing variables should be evaluated." Seems disconnect from the previous sentence. Please consider rephrasing it.

L451-454: the sentence was modified as follows:

*Old version:*

"On the other hand, the estimation of reservoirs' outflow using neural network models, such as the CLSTM model used here can also contain several limitations. With the choice of the forcing variables being pointed out by several authors as crucial for a successful model (ASCE, 1996; Maier et al., 2010; Dolling and Varas, 2002; Wu et al., 2014; Juan et al., 2017), the consideration of other forcing variables should be evaluated."

*New version:*

"On the other hand, the estimation of reservoirs' outflow using neural network models, such as the CLSTM model used here can also contain several limitations. As pointed out by several authors (ASCE, 1996; Maier et al., 2010; Dolling and Varas, 2002; Wu et al., 2014; Juan et al., 2017), the choice of the forcing variables

is a crucial task for a successful model. Thus, the consideration of other possible forcing variables for the CLSTM model should be evaluated."

L487: Did you mean by "environmentally friendly streamflow" the "environmental/ecological flow"? If yes, please rephrase it.

L487: Done.

**5 Conclusion**

L529: As you did not do the test I would consider changing this word from "…in part…" to "…probably…".

L529: Consider the inclusion of: "… on the performance of the coupled system in the computation of daily streamflow…".

L529: Done.

**References:**

Oliveira, A. R., Ramos, T. B., Simionesei, L., Pinto, L., and Neves, R.: Sensitivity Analysis of the MOHID-Land Hydrological Model: A Case Study of the Ulla River Basin. Water, 12(11), 3258, https://doi.org/10.3390/w12113258, 2020.

Moriasi DN, Gitau MW, Pai N, Daggupati P (2015) Hydrologic and water quality models: Performance measures and evaluation criteria. Trans ASABE (am Soc Agric Biol Eng) 58(6):1763–1785. https://doi.org/10.13031/trans.58.10715The complete list with the proposed technical corrections is attached in PDF.

---

## Author Comment (AC3)

Dear editor and reviewers,

Thank you for your constructive comments and suggestions about our manuscript. We revised the manuscript taking into account your suggestions and comments. Please find attached a point-by-point response, with our answers in blue. We hope that the revised version of the manuscript properly addresses your concerns.

**Sincerely,**

**Ana Oliveira on behalf of all authors**

**Referee #2 - Warrick Dawes**

egusphere-2023-915 "Direct integration of reservoirs' operations in a hydrological model for streamflow estimation: coupling a CLSTM model with MOHID-Land" AR.Oliveira, TB.Ramos, L.Pinto, R.Neves

The work presented in this article is very thorough. It is a valuable addition to the material from Oliveira et al. (2020) illustrating the use of AI/ML techniques, some of which were outlined in Oliveira et al. (2023) when applied to streamflow only. While there are a few peculiarities of wording through the text, the English expression for the most part is very good and it reads well.

We would like to thank referee #2 for the time spent evaluating our manuscript and the positive comments made to our work.

For corrective suggestions, perhaps only the line figures showing flow need to be cleaned up (Figures 2 and 7-9). With daily instantaneous data and small dots connected by lines, the hydrograph becomes a red sludge with the occasional peak that is visually unsatisfactory. Perhaps weekly or monthly volumetric totals would be more distinct, or single years shown as examples of the best/worst fit for the particular solution. This does not mean losing any of the finer daily detail when reporting statistics or minimum and maximum daily flows, as with the current tables.

The original figures will be replaced by new figures where the monthly values are presented, as suggested by the referee. Thus, Figure 2 will be replaced by:

Figure 2 Comparison of inflow and outflow volumes in (a) Portodemouros, (b) Touro, and (c) Bandariz reservoirs for the period 2010-2018, and in (d) Portodemouros reservoir for the period 1990-2018.

And Figure 7-9 will be replaced by:

Figure 7 Comparison of modelled and observed average monthly streamflow in hydrometric stations (a) Ulla-Touro and (b) Ulla-Teo with and without considering the existence of reservoirs. Focus on the daily values for the period between September 2013 and September 2014 in (c) Ulla-Touro and (d) Ulla-Teo hydrometric stations.